# SELF-SUPERVISED REPRESENTATION LEARNING VIA LATENT GRAPH PREDICTION

## ABSTRACT

Self-supervised learning (SSL) of graph neural networks is emerging as a promising way of leveraging unlabeled data. Currently, most methods are based on contrastive learning adapted from the image domain, which requires view generation and a sufficient number of negative samples. In contrast, existing predictive models do not require negative sampling, but lack theoretical guidance on the design of pretext training tasks. In this work, we propose the *LaGraph*, a theoretically grounded predictive SSL framework based on latent graph prediction. Learning objectives of *LaGraph* are derived as self-supervised upper bounds to objectives for predicting unobserved latent graphs. In addition to its improved performance, *LaGraph* provides explanations for recent successes of predictive models that include invariance-based objectives. We provide theoretical analysis comparing *LaGraph* to related methods in different domains. Our experimental results demonstrate the superiority of *LaGraph* in performance and the robustness to decreasing of training sample size on both graph-level and node-level tasks.

## 1 INTRODUCTION

Self-supervised learning (SSL) methods seek to use supervisions provided by data itself and design effective pretext learning tasks. These methods allow deep models to learn from a massive amount of unlabeled data and have achieved promising successes in natural language processing (Devlin et al., 2019; Wu et al., 2019; Wang et al., 2019) and image tasks (Batson and Royer, 2019; Xie et al., 2020; He et al., 2020; Chen et al., 2020). To use unlabeled graph data, earlier studies (Perozzi et al., 2014a; Grover and Leskovec, 2016) adapt sequence-based SSL methods (Mikolov et al., 2013b;a) to learn node representations. Inspired by the recent success of SSL in the image domain, a variety of SSL methods based on graph neural networks (GNNs) have been proposed in different learning paradigms. In particular, (Veličković et al., 2019; Zhu et al., 2020; Thakoor et al., 2021; Hassani and Khasahmadi, 2020; You et al., 2020) construct SSL tasks as unsupervised approaches to learn representations from graph data at either node-level or graph-level; (Hu et al., 2020) proposes SSL strategies to pre-train GNNs for downstream tasks; and (Jin et al., 2020; Kim and Oh, 2021) employ SSL as auxiliary tasks to boost the performance of main learning tasks.

Common taxonomies in recent survey works (Xie et al., 2021; Liu et al., 2020) consider two categories of SSL methods to train GNNs; namely, contrastive methods and predictive methods. Contrastive methods employ pair-wise discrimination as their pretext learning tasks. It performs transformations or augmentations to obtain multiple views from a graph and trains GNNs to discriminate between jointly sampled view pairs and independently sampled view pairs. In contrast, predictive methods (Hamilton et al., 2017; Hwang et al., 2020; Rong et al., 2020) train GNNs to predict certain labels obtained from the input graph, such as node reconstruction, connectivity reconstruction, graph statistical properties, and domain knowledge-based targets.

Adapted from the image domain, current state-of-the-art SSL methods for graphs are mostly contrastive. As a drawback, they usually depend on large training sample size to include a sufficient number of negative samples. With limited computing resources, contrastive methods may not be applicable to large-scale graphs without suffering from performance loss. To address the drawback, BGRL (Thakoor et al., 2021) adapts BYOL (Grill et al., 2020) to the graph domain. BGRL still obtains different views from each given graph, but it eliminates the requirement of negative samples by replacing contrastive objectives with the prediction of offline embedding. BGRL has achieved

competitive performance to the contrastive methods. However, unlike contrastive methods grounded by mutual information estimation and maximization, BYOL and BGRL lack theoretical guidance and require implementation measures to prevent collapsing to trivial representations, such as stop gradient, EMA, and normalization layers.

In this work, we propose *LaGraph*, a predictive SSL framework for representation learning of graph data, based on self-supervised latent graph prediction. In particular, we describe the notion of the latent graph and introduce the latent graph prediction as a pretext learning task. We adapt the supervised objective of latent graph prediction into a self-supervised setting by deriving its self-supervised upper bounds, according to which we present the learning framework of *LaGraph*. We provide further justifications of *LaGraph* by comparing it with theoretically sound methods in different domains. Our experimental results demonstrate the effectiveness of *LaGraph* on both graph-level and node-level representation learning, where a remarkable performance boost is achieved on a majority of datasets with higher stability to smaller batch size or training on subsets of nodes.

**Relations with Prior Work:** Both *LaGraph* and some existing contrastive methods (You et al., 2020; Zhu et al., 2020; Hu et al., 2020) apply node masking. A major difference is that *LaGraph* is a predictive method instead of contrastive. While those contrastive methods use node masking as an augmentation to obtain different views for contrast, *LaGraph* employ it for the computation of the invariance regularization. In addition, the objective of BGRL can be considered similar to the invariance regularization term in our objective. While the objective of BGRL is designed without theoretical guidance, our derived theorems associated with *LaGraph* objectives can explain the success of BGRL to some extent and provide guidance on how to better adopt objectives related to the invariance regularization on graphs.

## 2 METHODS

### 2.1 NOTATIONS AND PROBLEM FORMULATION

We consider an undirected graph $G = (V, E)$ with a set of attributed nodes $V$ and a set of edges $E$. We formulate the graph data as a tuple of matrices $(\boldsymbol{A}, \boldsymbol{X})$, where $\boldsymbol{A} \in \mathbb{R}^{|V| \times |V|}$ denotes the adjacency matrix and $\boldsymbol{X} \in \mathbb{R}^{|V| \times d}$ denotes the node features of dimension $d$. We employ a graph encoder $\mathcal{E}$ based on graph neural networks (GNNs) to encode each node or graph into a corresponding representation. Namely, we compute the node-level representations or node embedding by $\boldsymbol{H} = \mathcal{E}(\boldsymbol{A}, \boldsymbol{X}) \in \mathbb{R}^{|V| \times q}$ and the graph-level representation or graph embedding by $\boldsymbol{z} = \mathcal{R}(\boldsymbol{H}) \in \mathbb{R}^{1 \times q}$, where $q$ denotes the embedding dimension and $\mathcal{R} : \mathbb{R}^{|V| \times q} \to \mathbb{R}^{1 \times q}$ is a readout function.

Self-supervised representation learning is employed to train the graph encoder $\mathcal{E}$ on a set of $K$ graphs $\{G_i\}_{i=1}^{K}$ without labels from downstream tasks. In particular, we seek to design effective pre-text learning tasks, whose labels are obtained by task designation or from given data, to train the graph encoder $\mathcal{E}$ and produces informative representations for downstream tasks. Depending on the pre-text learning tasks, the encoder $\mathcal{E}$ is usually trained together with some prediction head $\mathcal{D}$ for predictive SSL or a discriminator for contrastive SSL.

### 2.2 LATENT GRAPH PREDICTION

Our method considers latent graph prediction as a pretext task to train graph neural networks. In this subsection, we introduce the general notion of latent data, followed by its specific definition for graph data, and the construction of the learning task. For any observed data instance $\boldsymbol{x}$, we assume that there exists a corresponding latent data $\boldsymbol{x}_{\mathcal{I}}$, determining the semantic of $\boldsymbol{x}$, such that the latent data $\boldsymbol{x}_{\mathcal{I}}$ is generated from a prior $p(\boldsymbol{x}_{\mathcal{I}})$ and the observed data instance is further generated from a certain distribution conditioned on the latent data, *i.e.*, $p(\boldsymbol{x}|\boldsymbol{x}_{\mathcal{I}})$. The most common case for the pair of observed data and latent data is the noisy data and its clean version.

When it comes to graph data, we consider the case that an observed graph data $G = (\boldsymbol{A}, \boldsymbol{X})$ is (noisily) generated from its latent graph $G_{\ell} = (\boldsymbol{A}, \boldsymbol{F})$ with the same node set and edge set, where node feature matrices $\boldsymbol{X}$ and $\boldsymbol{F}$ for the two graphs have the same dimensionality. We make two assumptions to the graphs without loss of generality. First, we assume that the observed feature vector $\boldsymbol{x}_v$ of each node $v$ in an observed graph is independently generated from a certain distribution

conditioned on the corresponding latent graph. In other words, how $\boldsymbol{x}_v$ is generated from the latent feature $\boldsymbol{f}_v$ is not affected by the generation of other observed feature vectors. Second, we assume that the conditional distribution of the observed graph is centered at the latent graph, *i.e.*, $\mathbb{E}[\boldsymbol{X}|G_\ell] = \boldsymbol{F}$. The above assumptions are natural when we have little knowledge about the generation process and are commonly used in other types of data such as the non-structural and zero-mean noise in images. In cases that the generation processes of different nodes are related or the distribution is not centered at $F$, we can still consider the related or biased components into the latent feature and therefore have the assumptions satisfied.

As the latent data usually determine the semantic meaning of observed data, we believe the prediction of latent graph can provide informative supervision for the learning of both graph-level and node-level representations. We are hence interested in constructing the learning task of latent graph prediction. To perform latent graph prediction, it is straightforward to employ a graph neural network $f : \{0,1\}^{|V|\times|V|} \times \mathbb{R}^{|V|\times d} \to \mathbb{R}^{|V|\times d}$ that takes an observed graph $G = (\boldsymbol{A}, \boldsymbol{X})$ as inputs and predicts the feature matrix of its latent graph $G_{\mathcal{I}} = (\boldsymbol{A}, \boldsymbol{F})$. When the ground truth of the latent feature matrix $\boldsymbol{F}$ is known, the learning objective can be designed as

$$f^* = \arg\min_f \mathbb{E}\,\|f(\boldsymbol{A}, \boldsymbol{X}) - \boldsymbol{F}\|^2. \tag{1}$$

Intuitively, the latent graph prediction can be considered as a generalized task from noisy data reconstruction that predicts the signal from the noisy data with the objective $\arg\min_f \mathbb{E}\,\|f(\boldsymbol{x}) - \boldsymbol{s}\|^2$, where the mapping from the signal to the noisy data $p(\boldsymbol{x}|\boldsymbol{s})$ can usually be explicitly modeled and samples of signal (ground truth) can usually be captured. In the data reconstruction case, pairs of $(\boldsymbol{x}, \boldsymbol{s})$ can be therefore directly captured or synthetically generated given a certain noise model $p(\boldsymbol{x}|\boldsymbol{s})$. However, when the task is generalized to latent graph prediction, there is a key challenge preventing us from directly applying the prediction task. That is, whereas there are natural supervisions for noisy data reconstruction, the latent graph is not observed and we are unable to explicitly model the mapping from latent graphs to observed graphs, *i.e.*, the conditional distribution $p(G|G_{\mathcal{I}})$.

## 2.3 Self-Supervised Upper Bounds for Latent Graph Prediction

As discussed in the previous subsection, unlike typical noisy data reconstruction tasks, the latent graph is not observed and $p(G|G_{\mathcal{I}})$ cannot be modeled explicitly. This makes it difficult to construct a direct learning task for latent graph prediction using the objective in Equation (1). We therefore seek to optimize an alternative objective that approximately optimizes the objective in Equation (1) without requiring the distribution $p(G|G_{\mathcal{I}})$, nor features $\boldsymbol{F}$ of the latent graph. We now introduce the proposed self-supervised objective for latent graph prediction.

We derive our self-supervised objective without involving $\boldsymbol{F}$ by constructing an upper bound of the objective in Equation (1). Specifically, we let $J \subset \{0, \cdots, |V| - 1\}$ be an arbitrary subset of node indices, $J^c$ denote the complement of set $J$, and $\boldsymbol{X}_{J^c} := \mathbb{1}_{J^c} \odot \boldsymbol{X} + \mathbb{1}_J \odot \boldsymbol{M}$ be the feature matrix with features of nodes in $V_J$ masked, where $\odot$ denotes element-wise multiplication, $\boldsymbol{M} \in \mathbb{R}^{|V|\times d}$ denotes a matrix consisting of independent random noise or zeros as masking values, and $\mathbb{1}_J \in \mathbb{R}^{|V|\times d}$ denotes an indicator matrix such that $\mathbb{1}_J[i,:] = \boldsymbol{1}, \forall i \in J$ and $\mathbb{1}_J[i,:] = \boldsymbol{0}, \forall i \notin J$. We describe the self-supervised upper bound in Theorem 1, whose proof is provided in Appendix A.

**Theorem 1.** *Consider a graph $G = (\boldsymbol{A}, \boldsymbol{X})$ and its latent graph $G_{\mathcal{I}} = (\boldsymbol{A}, \boldsymbol{F})$. We let the variance of any elements in $\boldsymbol{X}$ be bounded by $\sigma^2$ and $J$ be a subset of nodes $V$ in the graph $G$. For any graph neural network $f : \{0,1\}^{|V|\times|V|} \times \mathbb{R}^{|V|\times d} \to \mathbb{R}^{|V|\times d}$, we have the following inequality*

$$\mathbb{E}_{\boldsymbol{A},\boldsymbol{X},\boldsymbol{F}}\left[\|f(\boldsymbol{A}, \boldsymbol{X}) - \boldsymbol{F}\|^2 + \|\boldsymbol{X} - \boldsymbol{F}\|^2\right] \leq \mathbb{E}_{\boldsymbol{A},\boldsymbol{X}}\|f(\boldsymbol{A}, \boldsymbol{X}) - \boldsymbol{X}\|^2 +$$

$$2\sigma|V|\,\mathbb{E}_J\left[\frac{\mathbb{E}_{\boldsymbol{A},\boldsymbol{X}}\|f_J(\boldsymbol{A}, \boldsymbol{X}) - f_J(\boldsymbol{A}, \boldsymbol{X}_{J^c})\|^2}{|J|}\right]^{1/2}. \tag{2}$$

Intuitively, the first component in the upper bound derived in Theorem 1 measures the reconstruction error on the feature matrix $\boldsymbol{X}$ of the given observed graph $G$, enforcing the intermediate representations to be informative. The second component controls how much information is accessible from the input feature of a node $v_i$ when reconstructing the feature of $v_i$, by encouraging the output of a

node to be invariant to the missing of its features in the input graph. We then call the first component as a reconstruction term and the second component as an invariance regularization term. Note that the invariance regularization is only computed on masked nodes in contrast to the BGRL objective, based on different theoretical grounding and leading to a different effect (see Appendix D).

In tasks of self-supervised representation learning, we are more interested in graph-level or node-level representations than predicted latent graphs. In these cases, we expect the representations also holds the invariance property held by the final outputs. We therefore seek to apply the invariance regularization to the representations, since a regularization applied to the output does not necessarily control the information accessibility of representations produced intermediately in the graph neural network. To do so, we separately consider the encoder $\mathcal{E}$ and decoder $\mathcal{D}$ in the graph neural network $f$. We introduce certain assumptions to the decoder network $\mathcal{D}$ and the readout function $\mathcal{R}$, and derive two additional upper bounds for node-level and graph-level representation learning, respectively in following corollaries. Proofs of the corollaries are provided in Appendix B.

**Corollary 1.** *Let $G = (\boldsymbol{A}, \boldsymbol{X})$ be a given graph, $G_{\mathcal{I}} = (\boldsymbol{A}, \boldsymbol{F})$ be its latent graph, $\mathcal{E}$ and $\mathcal{D}$ be a graph encoder and a prediction head (decoder) consisting of fully-connected layers. If the prediction head $\mathcal{D}$ is $\ell$-Lipschitz continuous with respect to $l_2$-norm, we further have the following inequality,*

$$\mathbb{E}\left[\|\mathcal{D}(\boldsymbol{H}) - \boldsymbol{F}\|^2 + \|\boldsymbol{X} - \boldsymbol{F}\|^2\right] \leq \mathbb{E}\|\mathcal{D}(\boldsymbol{H}) - \boldsymbol{X}\|^2 + 2\sigma|V|\ell\,\mathbb{E}_J\left[\frac{\mathbb{E}\|\boldsymbol{H}_J - \boldsymbol{H}'_J\|^2}{|J|}\right]^{1/2},$$

(3)

*where $\boldsymbol{H} = \mathcal{E}(\boldsymbol{A}, \boldsymbol{X})$ and $\boldsymbol{H}' = \mathcal{E}(\boldsymbol{A}, \boldsymbol{X}_{J^c})$ denote the node embedding of the given graph and the masked graph, respectively, and $\boldsymbol{H}_J := \boldsymbol{H}[J, :]$ selects rows with indices in $J$.*

**Corollary 2.** *Let $G = (\boldsymbol{A}, \boldsymbol{X})$ be a given graph, $G_{\mathcal{I}} = (\boldsymbol{A}, \boldsymbol{F})$ be its hidden latent graph, $\mathcal{E}$ be a graph encoder, $\mathcal{R}$ be a readout function satisfying $k$-Bilipschitz continuity with respect to $l_2$-norm, and $\mathcal{D}$ be a prediction head (decoder). If the prediction head $\mathcal{D}$ is $\ell$-Lipschitz continuous with respect to $l_2$-norm, we have the following inequality,*

$$\mathbb{E}\left[\|\mathcal{D}(\boldsymbol{H}) - \boldsymbol{F}\|^2 + \|\boldsymbol{X} - \boldsymbol{F}\|^2\right] \leq \mathbb{E}\|\mathcal{D}(\boldsymbol{H}) - \boldsymbol{X}\|^2 + 2\sigma|V|k\ell\,\mathbb{E}_J\left[\frac{\mathbb{E}\|\boldsymbol{z} - \boldsymbol{z}'\|^2}{|J|}\right]^{1/2}, \quad (4)$$

*where $\boldsymbol{z} = \mathcal{R}(\boldsymbol{H})$ and $\boldsymbol{z}' = \mathcal{R}(\boldsymbol{H}')$ denote the graph-level representations of the given graph and the masked graph, respectively.*

We note that the assumptions and restrictions are natural or practically satisfiable. The assumption that variance of each element in $\boldsymbol{X}$ is bounded by $\sigma$ holds when node features are from $\{0, 1\}^d$ or when feature normalization is applied. The $\ell$-Lipschitz continuous property is common for neural networks. And the $k$-Bilipschitz continuity can be satisfied by applying an injective readout function such as global sum pooling, which is commonly used in graph-level tasks.

## 2.4 THE *LaGraph* FRAMEWORK

We design our self-supervised learning framework according to upper bounds derived in Corollary 1 and Corollary 2. To train encoder $\mathcal{E}$ together with decoder $\mathcal{D}$ under self-supervision, we input to the encoder both the given graph $(\boldsymbol{A}, \boldsymbol{X})$ and its variation $(\boldsymbol{A}, \boldsymbol{X}_{J^c})$ with a random subset $J$ of node indices for nodes to be masked and obtain node-level representations $\boldsymbol{H} = \mathcal{E}(\boldsymbol{A}, \boldsymbol{X})$ and $\boldsymbol{H}' = \mathcal{E}(\boldsymbol{A}, \boldsymbol{X}_{J^c})$ for the two graphs respectively. The self-supervised losses are computed on input node features, reconstructed node features, and representations, as demonstrated in Figure 1.

In particular, we consider a mini-batch of $N$ graphs $\{(\boldsymbol{A}_i, \boldsymbol{X}_i)\}_{i=1}^N$ and their corresponding masked variation $\{(\boldsymbol{A}_i, \boldsymbol{X}_{(i,J_i^c)})\}_{i=1}^N$ where $J_i$ denotes the node indices subset for the $i$-th graph. The self-supervised loss for node-level representation learning follows Corollary 1 and is computed as

$$L_{node}(\mathcal{E}, \mathcal{D}) = \frac{1}{N}\sum_{i=1}^N \|\mathcal{D}(\boldsymbol{A}_i, \boldsymbol{H}_i) - \boldsymbol{X}_i\|^2 / |V_i| + \alpha\left[\frac{\sum_i \|\mathbb{1}_{J_i} \odot \boldsymbol{H}_i - \mathbb{1}_{J_i} \odot \boldsymbol{H}'_i\|^2}{\sum_i |J_i|}\right]^{1/2}, \quad (5)$$

where $\alpha$ is a hyper-parameter corresponding to the multiplier $2\sigma\ell$ in Corollary 1. To fulfill the conditions in Corollary 1, we employ fully-connected layers instead of graph convolutional layers in the decoder $\mathcal{D}$. The pseudo-code are provided in Appendix C.

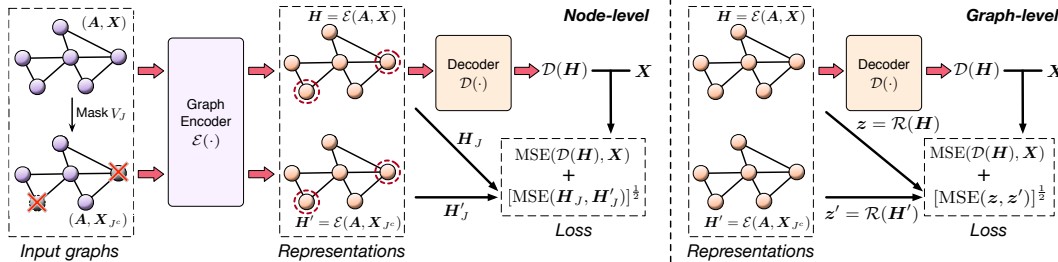

Figure 1: Overview of the *LaGraph* framework. Given a training graph, we randomly mask a small portion $V_J \in V$ of its nodes and input both original graph and masked graph to the encoder $\mathcal{E}$. Crossed nodes in the figure have all their attributes masked but topology preserved. The final loss consists of a reconstruction loss on node features and an invariance loss between representations of the original graph and the masked graph. We omit the encoding part of the graph-level framework as frameworks for the two levels mainly differ in whether the invariance term is computed on representations of masked nodes or graph-level representations obtained by $\mathcal{R}$.

Similarly, using the same notations above, the self-supervised loss for graph-level representation learning follows Corollary 2 and is computed as

$$L_{graph}(\mathcal{E}, \mathcal{D}) = \frac{1}{N} \sum_{i=1}^{N} \|\mathcal{D}(\boldsymbol{A}_i, \boldsymbol{H}_i) - \boldsymbol{X}_i\|^2 / |V_i| + \alpha' \left[ \sum_i \|\boldsymbol{z}_i - \boldsymbol{z}_i'\|^2 / \sum_i |J_i| \right]^{1/2}, \quad (6)$$

where $\boldsymbol{z}_i = \mathcal{R}(\boldsymbol{H}_i)$ and $\boldsymbol{z}_i' = \mathcal{R}(\boldsymbol{H}_i')$ denote the graph-level representations obtained by applying readout function $\mathcal{R}$ to the node-level representations, respectively, and $\alpha'$ is a hyper-parameter corresponding to the multiplier $2\sigma k\ell$ in Corollary 2. To fulfill the conditions in Corollary 2, we employ global sum pooling as the readout function $\mathcal{R}$, where as the decoder $\mathcal{D}$ here can consists of either fully-connected layers or graph convolutional layers. The pseudo-code are provided in Appendix C.

# 3    THEORETICAL ANALYSIS AND RELATIONS WITH PRIOR WORK

In this section, we further theoretically justify and motivate *LaGraph* by providing comparisons and connections between our method and existing related methods, including denoising autoencoders (Vincent et al., 2010; Wang et al., 2017), information bottleneck principle (Tishby et al., 1999), and contrastive methods based on local-global mutual information maximization (Veličković et al., 2019; Sun et al., 2019; Hassani and Khasahmadi, 2020).

## 3.1    DENOISING AUTOENCODERS

Denoising autoencoders employ an encoder-decoder network architecture and perform self-supervised training by masking or corrupting a portion of dimensions of the given data, and reconstruct the masked or corrupted value given their context. Such an approach have been also applied for self-supervised image denoising (Batson and Royer, 2019), known as the blind-spot denoising. Similarly to our method, the denoising autoencoder can be also viewed as an approximation of the latent graph prediction. Using the same notation in Section 2, we formulate the connection between latent graph prediction and the graph denoising autoencoder in the following theorem.

**Theorem 2.** *Let $J$ be a uniformly sampled subset of node indices of the given graph $(\boldsymbol{A}, \boldsymbol{X})$, $\mathcal{F}$ be the class of all graph neural networks, and $\mathcal{F}^*$ be the class of graph neural networks such that $f_J^*(\boldsymbol{A}, \boldsymbol{X})$ does not depend on $\boldsymbol{X}_J$, for any $J$ and $f^* \in \mathcal{F}^*$. Given any graph neural network $f \in \mathcal{F}$, there exist $f^* \in \mathcal{F}^*$ and $f' \in \mathcal{F}$ such that*

$$\mathbb{E}_{\boldsymbol{A}, \boldsymbol{X}, \boldsymbol{F}} \left[ \|f(\boldsymbol{A}, \boldsymbol{X}) - \boldsymbol{F}\|^2 + \|\boldsymbol{X} - \boldsymbol{F}\|^2 \right] = \mathbb{E}_{\boldsymbol{A}, \boldsymbol{X}} \|f(\boldsymbol{A}, \boldsymbol{X}) - \boldsymbol{X}\|^2 +$$

$$\mathbb{E}_{\boldsymbol{A}, \boldsymbol{X}, \boldsymbol{F}} \left[ 2 \langle f(\boldsymbol{A}, \boldsymbol{X}) - \boldsymbol{F}, \boldsymbol{X} - \boldsymbol{F} \rangle \right] \quad (7)$$

$$\approx \mathbb{E}_{\boldsymbol{A}, \boldsymbol{X}} \|f^*(\boldsymbol{A}, \boldsymbol{X}) - \boldsymbol{X}\|^2 \quad (8)$$

$$= |V| \mathbb{E}_J \mathbb{E}_{\boldsymbol{A}, \boldsymbol{X}} \|f_J'(\boldsymbol{A}, \boldsymbol{X}_{J^c}) - \boldsymbol{X}_J\|^2 / |J|. \quad (9)$$

Equation (7) is proved in the proof of Theorem 1. It can be verified that the second term, *i.e.*, the expectation of the inner product, in Equation (7) reduces to zero when the neural network $f$ satisfies that $f_J(\boldsymbol{A}, \boldsymbol{X})$ does not depend on $\boldsymbol{X}_J$, for any $J$, according to Batson and Royer (2019). The objective can be therefore approximated by Equation (8) with the neural network $f^*$ satisfying such a property. To let any graph neural network $f$ satisfy the property, one can apply masks to a portion of nodes indexed by $J$ so that their original value is inaccessible by $f$ when predicting $f_J(\boldsymbol{A}, \boldsymbol{X})$. Therefore, the latent graph prediction objective under supervision can be further approximated by Equation (9), who describes the objective of a graph denoising autoencoder.

A substantial difference between our method and the denoising autoencoder lies in how to handle the inner product term in Equation (7). In particular, the denoising autoencoder forces the term to be zero by assuming certain property of the graph neural network, whereas our method derives an upper bound, *i.e.*, the invariance term, for the inner product. Theoretically, the graph denoising autoencoder is equivalent to our framework with infinite weight scalar for the invariance term. As a drawback, when $f_J(\boldsymbol{A}, \boldsymbol{X})$ does not depend on $\boldsymbol{X}_J$, the learned representations can be less informative as representations of nodes in $V_J$ do not include the information of $\boldsymbol{X}_J$, for any $J$, leading to performance loss. Our proposed upper bounds allow an encoder to access a certain level of information of the masked nodes, whose representations can be as good as ones from supervised learning. In fact, our method can be viewed as an autoencoder with an invariance regularization.

## 3.2 THE INFORMATION BOTTLENECK PRINCIPLE

The information bottleneck principle (Tishby et al., 1999) is a technique for data compression and signal processing in the field of information theory, and has been widely applied in deep learning problems (Tishby and Zaslavsky, 2015; Saxe et al., 2018). Let $\boldsymbol{X}$ be a random variable to be compressed, $\tilde{\boldsymbol{X}}$ be an observed relevant variable, and $\boldsymbol{Z}$ denote the compressed representation of $\boldsymbol{X}$. The information bottleneck principle seeks to optimize the following problem

$$\boldsymbol{Z}^* = \arg\min_{\boldsymbol{Z}} I(\boldsymbol{Z}; \tilde{\boldsymbol{X}}) - \beta I(\boldsymbol{Z}; \boldsymbol{X}), \tag{10}$$

where $I(\cdot; \cdot)$ denotes the mutual information and $\beta > 1$ is a Lagrange multiplier. The work Barlow Twin (Zbontar et al., 2021) has discussed a connection between the information bottleneck principle and self-supervised learning. In particular, to apply information bottleneck to SSL, one usually obtain $\tilde{\boldsymbol{X}}$ by performing augmentations or distortions on the given data $\boldsymbol{X}$. And Equation (10) can be rewritten into

$$\boldsymbol{Z}^* = \arg\min_{\boldsymbol{Z}} \left[ H(\boldsymbol{Z}) - H(\boldsymbol{Z}|\tilde{\boldsymbol{X}}) \right] - \beta \left[ H(\boldsymbol{Z}) - H(\boldsymbol{Z}|\boldsymbol{X}) \right] \tag{11}$$

$$= \arg\min_{\boldsymbol{Z}} H(\boldsymbol{Z}|\boldsymbol{X}) - \lambda H(\boldsymbol{Z}), \tag{12}$$

where $\lambda = \frac{\beta - 1}{\beta} > 0$ is a weight scalar. Intuitively, the conditional entropy $H(\boldsymbol{Z}|\boldsymbol{X})$ is to be minimized, indicating that the distortion should add no additional information to the representation $\boldsymbol{Z}$. In other words, the representation $\boldsymbol{Z}$ should be as invariant as possible to distortions applied to $\boldsymbol{X}$. In addition, the entropy $H(\boldsymbol{Z})$ is to be maximized, indicating that the representation $\boldsymbol{Z}$ itself should be as informative as possible.

The two terms in objectives of *LaGraph* correspond to the terms in Equation (12). In particular, the invariance term corresponding to $H(\boldsymbol{Z}|\boldsymbol{X})$ and the reconstruction term aims to ensure informative representations, *i.e.*, to maximize $H(\boldsymbol{Z})$. Objectives in existing SSL methods such as BYOL (Grill et al., 2020), its variation BGRL (Thakoor et al., 2021) in graph domain, and Barlow Twin (Zbontar et al., 2021) also include invariance terms corresponding to $H(\boldsymbol{Z}|\boldsymbol{X})$. To encourage informative representations, Barlow Twin further include a redundancy reduction term to minimize the cross-correlation between different dimensions of the representation, as a proxy of the maximization of $H(\boldsymbol{Z})$. In spite of the similarity, there are still substantial differences as discussed in Appendix D. In addition, the InfoNCE (NT-XENT) loss employed in some contrastive learning methods (You et al., 2020; Zhu et al., 2020) induces a similar effect, according to Zbontar et al. (2021). Both Equation (12) and the derivation of *LaGraph* objectives indicate the importance of the invariance term in SSL objectives. In addition, compared to the redundancy reduction term in Barlow Twin and the noise contrast in InfoNCE, *LaGraph* objectives can directly guarantee the learning of informative representations measured by the reconstruction capability.

### 3.3 Contrastive Learning by Maximizing Local-Global Mutual Information

Motivated by Deep InfoMax (Hjelm et al., 2019), recent graph self-supervised learning methods (Veličković et al., 2019; Sun et al., 2019; Hassani and Khasahmadi, 2020) constructs their learning tasks by maximizing the mutual information between local (node-level) representations and a global (graph-level) summary of the graph. Practically, as a $k$-layer encoder $\mathcal{E}$ has the receptive field of at most $k$-hop neighborhood, the goal becomes the maximization of the mutual information between local representations and their $k$-hop neighborhood, formulated as

$$\mathcal{E}^* = \arg \max_{\mathcal{E}} \sum_{i=1}^{|V|} I(\boldsymbol{X}_i^{(k)}; \mathcal{E}_i(\boldsymbol{A}, \boldsymbol{X})), \qquad (13)$$

where $I$ denotes the mutual information, $\boldsymbol{X}_i^{(k)}$ is the $k$-hop neighborhood of node $i$, $\mathcal{E}$ is a graph encoder with $k$ GNN layers, and $\mathcal{E}_i(\boldsymbol{A}, \boldsymbol{X})$ denotes the local representation of node $i$. The learning objective is motivated by the goal that the local representations should contain as much the global information of the entire graph (or the $k$-hop neighborhood) as possible.

As for *LaGraph*, the reconstruction term encourages representations to contain sufficient information to reconstruct the input features while the invariance term limits the information accessibility from a local node when reconstructing its features. The two terms in the objective jointly promote node representations to learn limited local information and as much contextual information from the neighborhood as possible for reconstruction. It hence has a similar effect to the local-global mutual information maximization.

## 4 Experiments

We conduct experiments on both node-level and graph-level self-supervised representation learning tasks with datasets used in two most recent state-of-the-art methods for SSL (You et al., 2020; Thakoor et al., 2021). For graph-level tasks, we follow GraphCL (You et al., 2020) to perform evaluations on eight graph classification datasets (Wale and Karypis, 2006; Borgwardt et al., 2005; Dobson and Doig, 2003; Debnath et al., 1991; Yanardag and Vishwanathan, 2015) from TUDataset (Morris et al., 2020). For node-level tasks, as the citation network datasets (McCallum et al., 2000; Giles et al., 1998; Sen et al., 2008) are recognized to be saturated and unreliable for GNN evaluation (Shchur et al., 2018; Thakoor et al., 2021), we follow Thakoor et al. (2021) to include four transductive node classification datasets from (Shchur et al., 2018), including Amazon Computers, Amazon Photos from the Amazon Co-purchase Graph (McAuley et al., 2015), Coauthor CS, and Coauthor Physics from the Microsoft Academic Graph (Sinha et al., 2015). We further include three larger scale inductive datasets, PPI, Reddit, and Flickr, for node-level classification used in SUBG-CON (Jiao et al., 2020).

We follow You et al. (2020) and Zhu et al. (2020) for the standard linear evaluation protocols at graph-level and node-level, respectively. In particular, for both levels, we first train the graph encoder on unlabeled graph datasets with the corresponding self-supervised objective. We then compute and freeze the corresponding representations and train a linear classification model on top of the fixed representations with their corresponding labels. Linear SVM and the regularized logistic regression are employed as linear classifiers for graph-level datasets and node-level datasets, according to You et al. (2020) and Zhu et al. (2020), respectively. For inductive node-level datasets, the self-supervised training is only performed on graphs in the training datasets whereas the test graphs are unavailable during the self-supervised training.

### 4.1 Comparisons with Baselines

We perform experiments on both graph-level and node-level datasets to demonstrate the effectiveness of *LaGraph*. We construct our model and losses according to Section 2.4. Detailed model configurations, training settings, and dataset statistics are provided in Appendix G.

**Graph-level Datasets.** We evaluate the performance of *LaGraph* in terms of the linear classification accuracy and compare it with three kernel-based methods including graphlet kernel (GL) (Shervashidze et al., 2009), Weisfeiler-Lehman kernel (WL) (Shervashidze et al., 2011), and deep graph

Table 1: Performance on graph-level classification tasks, scores are averaged over 5 runs. Bold and underlined numbers highlight the top-2 performance. OOM indicates running out-of-memory on a 56GB Nvidia A6000 GPU.

|  | NCI1 | PROTEINS | DD | MUTAG | COLLAB | RDT-B | RDT-M5K | IMDB-B |
|---|---|---|---|---|---|---|---|---|
| GL | – | – | – | 81.7±2.1 | – | 77.3±0.2 | 41.0±0.2 | 65.9±1.0 |
| WL | 80.0±0.5 | 72.9±0.6 | – | 80.7±3.0 | – | 68.8±0.4 | 46.1±0.2 | 72.3±3.4 |
| DGK | 80.3±0.5 | 73.3±0.8 | – | 87.4±2.7 | – | 78.0±0.4 | 41.3±0.2 | 67.0±0.6 |
| Node2Vec | 54.9±1.6 | 57.5±3.6 | – | 72.6±10.2 | – | – | – | – |
| Sub2Vec | 52.8±1.5 | 53.0±5.6 | – | 61.1±15.8 | – | 71.5±0.4 | 36.7±0.4 | 55.3±1.5 |
| Graph2Vec | 73.2±1.8 | 73.3±2.1 | – | 83.2±9.3 | – | 75.8±1.0 | 47.9±0.3 | 71.1±0.5 |
| GAE | 73.3±0.6 | 74.1±0.5 | 77.9±0.5 | 84.0±0.6 | 56.3±0.1 | 74.8±0.2 | 37.6±1.6 | 52.1±0.2 |
| VGAE | 73.7±0.3 | 74.0±0.5 | 77.6±0.4 | 84.4±0.6 | 56.3±0.0 | 74.8±0.2 | 39.1±1.6 | 52.1±0.2 |
| InfoGraph | 76.2±1.1 | 74.4±0.3 | 72.9±1.8 | 89.0±1.1 | 70.7±1.1 | 82.5±1.4 | 53.5±1.0 | 73.0±0.9 |
| GraphCL | 77.9±0.4 | 74.4±0.5 | **78.6±0.4** | 86.8±1.3 | 71.4±1.2 | 89.5±0.8 | 56.0±0.3 | 71.1±0.4 |
| MVGRL | 75.1±0.5 | 71.5±0.3 | OOM | 89.7±1.1 | OOM | 84.5±0.6 | OOM | **74.2±0.7** |
| LaGraph | **79.9±0.5** | **75.2±0.4** | 78.1±0.4 | **90.2±1.1** | **77.6±0.2** | **90.4±0.8** | **56.4±0.4** | 73.7±0.9 |

Table 2: Performance on node-level datasets, 20 runs averaged. Results of SSL methods with the best performance are highlighted in bold numbers. *Left*: Mean classification accuracy on transductive datasets. *Right*: Micro-averaged F1 scores on larger-scale inductive datasets .

| Transductive | Am.Comp. | Am.Pht. | Co.CS | Co.Phy | Inductive | PPI | Flickr | Reddit |
|---|---|---|---|---|---|---|---|---|
| Raw features | 73.8±0.0 | 78.5±0.0 | 90.4±0.0 | 93.6±0.0 | Raw feat. | 42.5±0.3 | 20.3±0.2 | 58.5±0.1 |
| DeepWalk | 85.7±0.1 | 89.4±0.1 | 84.6±0.2 | 91.8±0.2 | GAE | 75.7±0.0 | 50.7±0.2 | OOM |
| GAE | 87.7±0.3 | 92.7±0.3 | 92.4±0.2 | 95.3±0.1 | VGAE | 75.8±0.0 | 50.4±0.2 | OOM |
| VGAE | 88.1±0.3 | 92.8±0.3 | 92.5±0.2 | 95.3±0.1 | Super-GCN | 51.5±0.6 | 48.7±0.3 | 93.3±0.1 |
| Supervised | 86.5±0.5 | 92.4±0.2 | 93.0±0.3 | 95.7±0.2 | Super-GAT | 97.3±0.2 | OOM | OOM |
| DGI | 84.0±0.5 | 91.6±0.2 | 92.2±0.6 | 94.5±0.5 | GraphSAGE | 46.5±0.7 | 36.5±1.0 | 90.8±1.1 |
| GMI | 82.2±0.3 | 90.7±0.2 | OOM | OOM | DGI | 63.8±0.2 | 42.9±0.1 | 94.0±0.1 |
| MVGRL | 87.5±0.1 | 91.7±0.1 | 92.1±0.1 | 95.3±0.0 | GMI | 65.0±0.0 | 44.5±0.2 | 95.0±0.0 |
| GRACE | 87.5±0.2 | 92.2±0.2 | 92.9±0.0 | 95.3±0.0 | SUBG-CON | 66.9±0.2 | 48.8±0.1 | **95.2±0.0** |
| GCA | 88.9±0.2 | 92.5±0.2 | 93.1±0.0 | 95.7±0.0 | BGRL-GCN | 69.6±0.2 | – | – |
| BGRL | **89.7±0.3** | 92.9±0.3 | 93.2±0.2 | 95.6±0.1 | BGRL-GAT | 70.5±0.1 | – | – |
| LaGraph | 88.0±0.3 | **93.5±0.4** | **93.3±0.2** | **95.8±0.1** | LaGraph | **74.6±0.0** | **51.3±0.1** | **95.2±0.0** |

kernel (DGK) (Yanardag and Vishwanathan, 2015), together with five unsupervised methods including Node2Vec (Grover and Leskovec, 2016), Sub2Vec (Adhikari et al., 2018), Graph2Vec (Narayanan et al., 2017), GAE and VGAE (Kipf and Welling, 2016). We further compare the results with recent SOTA SSL methods based on contrastive learning, including InfoGraph (Sun et al., 2019) , MVGRL (Hassani and Khasahmadi, 2020), and GraphCL (You et al., 2020). Results in Table 1 show that *LaGraph* outperforms the current SOTA methods on a majority of datasets and is on par with the best performance on the rest of datasets. Additional results adopting *LaGraph* as a pre-training strategy under the semi-supervised learning setting are provided in Appendix F.

**Node-level Datasets.** We perform node-level experiments on both transductive and inductive learning tasks. For the evaluation of transductive learning, we compare the performance of *LaGraph* in terms of linear classification accuracy with DeepWalk (Perozzi et al., 2014b), GAE, VGAE, and six contrastive learning methods including Deep Graph InfoMax (DGI) (Veličković et al., 2019), GMI (Peng et al., 2020), MVGRL (Hassani and Khasahmadi, 2020), GRACE (Zhu et al., 2020), GCA (Zhu et al., 2021), and BGRL (Thakoor et al., 2021), where BGRL is the current state-of-the-art SSL method for node-level representation learning. We further include the results of directly performing linear classification on raw node feature (raw features) and by supervised training for references. To be consistent to Thakoor et al. (2021), we have ensured that the GPU memory consumption of *LaGraph* are under 16GB for the four transductive datasets. We then perform additional experiments on the larger-scale inductive datasets (Zitnik and Leskovec, 2017; Zeng et al., 2019; Hamilton et al., 2017) and compare our results in terms of micro-averaged F1-score with DeepWalk, unsupervised GraphSAGE (Hamilton et al., 2017), DGI, GMI, SUBG-CON (Jiao et al., 2020) and BGRL. Results for both transductive datasets and inductive datasets shown in Table 2 suggest competitive performance of *LaGraph* compared to the existing SOTA methods.

Table 3: Model performance when trained on a subset of nodes.

|  | # nodes sampled | 100 | 1,000 | 2,500 | 5,000 | 10,000 | all |
|---|---|---|---|---|---|---|---|
| Flickr | % nodes sampled | 0.22% | 2.24% | 5.60% | 11.20% | 22.41% | 100.00% |
|  | F1-score - *LaGraph* | 6.07 | 51.12 | 51.12 | 51.27 | 51.29 | 51.26 |
|  | Memory - *LaGraph* | 1389MB | 1465MB | 1553MB | 1725MB | 2065MB | 4211MB |
|  | F1-score - GraphCL | 45.27 | 45.27 | 45.27 | 45.38 | 45.45 | 45.48 |
|  | Memory - GraphCL | 1647MB | 2599MB | 4137MB | 6741MB | 11905MB | 47939MB |
| Reddit | % nodes sampled | 0.07% | 0.65% | 1.63% | 3.25% | 6.50% | 100.00% |
|  | F1-score - *LaGraph* | 5.76 | 95.05 | 95.06 | 95.08 | 95.09 | 95.22 |
|  | Memory - *LaGraph* | 1403MB | 1475MB | 1585MB | 1783MB | 2161MB | 16933MB |
|  | F1-score - GraphCL | 93.24 | 93.24 | 93.25 | 93.31 | 93.32 | OOM |
|  | Memory - GraphCL | 4199MB | 6117MB | 6687MB | 9297MB | 14495MB | OOM |

## 4.2 ABLATION STUDY

We further conduct three ablation studies to explore model robustness to smaller batch sizes on graph-level data and to training with sub-graphs on large-scale node-level datasets. An additional ablation study on the effect of optimizing different objectives is provided in Appendix G.

**Robustness to Batch Sizes.** Different from contrastive learning methods, *LaGraph* does not requires negative samples to perform noise contrast or pair-wise discrimination. Therefore, an advantage of *LaGraph* is that the performance is robust to the batch size as it does not depend on large batch sizes with sufficient negative samples. To verify the statement, we perform ablation study on how model performance changes when decreasing the batch size from 128 to 8 for graph-level datasets. We include corresponding results of GraphCL who uses InfoNCE for references and show the comparisons in Figure 2. The results indicate while contrastive methods based on InfoNCE suffer from significant performance loss with a small batch size, *LaGraph* are more robust to the batch size.

**Training on Sub-Graphs for Large-Scale Datasets.** Training graph encoders on all nodes for some large-scale graphs can be heavily expensive in computation. We hence conduct ablation study on how training graph encoders on a portion of sampled nodes instead of entire graph affect the effectiveness of training. Results in Table 3 suggest that the model performance remains stable when decreasing the number of nodes until the number becomes extremely

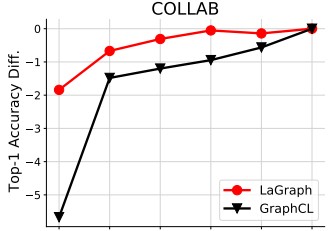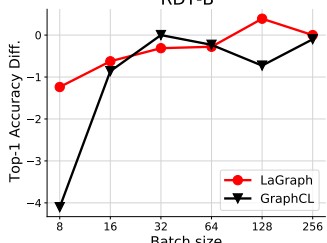

Figure 2: Model robustness to small batch sizes. Shown are changes in accuracy compared to the batch size of 256.

small. The collapse is due to the very sparse connectivity and *LaGraph* fails to reconstruct a node from its neighbor nodes as there is no neighbors at all. In contrast, though GraphCL does not collapse at extremely small subsets, it suffer more from performance loss above 1,000 nodes and consumes significantly more GPU memory.

## 5 CONCLUSIONS AND FUTURE DIRECTIONS

We introduced *LaGraph*, a state-of-the-art predictive SSL framework whose objectives are based on the self-supervised latent graph prediction. We provided theoretical analysis and discussed the relationship between *LaGraph* and theories in different related domains. Experimental results demonstrate the strong effectiveness of the proposed framework and the stability to the training scale for both graph-level and node-level tasks. Currently, our framework mainly considers the latent graph regarding its node features. Only limited topology information can be included by augmenting node features, *e.g.*, with node degrees. Further investigation into a latent graph prediction framework that includes richer information such as edge features and latent connectivity into self-supervision can potentially bring additional improvement on the performance. See more discussions in Appendix G.

## REPRODUCIBILITY STATEMENT

The experiment setting and evaluation protocol are described in Section 4. To ensure reproducibility, we provide implementation details, model configurations, and the selection of hyper-parameters in Appendix G. In addition, we provide the proof to all theorems and corollaries in Appendix A and Appendix B. The code to fully reproduce the results will be released upon acceptance of the paper.

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

## A    PROOF OF THEOREM 1

*Proof.* We first derive the relationship between the supervised objective of latent graph prediction $\mathbb{E}\|f(\boldsymbol{A}, \boldsymbol{X}) - \boldsymbol{F}\|^2$ and the self-supervised reconstruction loss $\mathbb{E}\|f(\boldsymbol{A}, \boldsymbol{X}) - \boldsymbol{X}\|^2$ in the following equations,

$$\mathbb{E}\|f(\boldsymbol{A}, \boldsymbol{X}) - \boldsymbol{X}\|^2 = \mathbb{E}\|(f(\boldsymbol{A}, \boldsymbol{X}) - \boldsymbol{F}) - (\boldsymbol{X} - \boldsymbol{F})\|^2 \tag{14}$$

$$= \mathbb{E}\big[\|f(\boldsymbol{A}, \boldsymbol{X}) - \boldsymbol{F}\|^2 + \|\boldsymbol{X} - \boldsymbol{F}\|^2 - 2\langle f(\boldsymbol{A}, \boldsymbol{X}) - \boldsymbol{F}, \boldsymbol{X} - \boldsymbol{F}\rangle\big], \tag{15}$$

where $\langle \cdot, \cdot \rangle$ denotes the inner product along all dimensions $\{0, \cdots, d|V| - 1\}$. The expectation $\mathbb{E}\|f(\boldsymbol{A}, \boldsymbol{X})\|$ in the above equation is not relevant to the neural network $f$. It hence can be considered as a constant during the optimization of $f$. To derive an upper bound to $\mathbb{E}\|f(\boldsymbol{A}, \boldsymbol{X}) - \boldsymbol{F}\|^2$, we only need to derive an upper bound of its equivalent $\mathbb{E}\|f(\boldsymbol{A}, \boldsymbol{X}) - \boldsymbol{X}\|^2 + 2\mathbb{E}\langle f(\boldsymbol{A}, \boldsymbol{X}) - \boldsymbol{F}, \boldsymbol{X} - \boldsymbol{F}\rangle$. As $\boldsymbol{F}$ is unobserved, our goal is to derive an upper bound to eliminate the need of $\boldsymbol{F}$ for the inner product term $\langle f(\boldsymbol{A}, \boldsymbol{X}) - \boldsymbol{F}, \boldsymbol{X} - \boldsymbol{F}\rangle$. To do so, we apply the definition of latent graph $\mathbb{E}[\boldsymbol{X}|\boldsymbol{A}, \boldsymbol{F}] = \boldsymbol{F}$ and rewrite the inner product into the following form.

$$\mathbb{E}\langle f(\boldsymbol{A}, \boldsymbol{X}) - \boldsymbol{F}, \boldsymbol{X} - \boldsymbol{F}\rangle = \mathbb{E}_{\boldsymbol{A}, \boldsymbol{F}} \mathbb{E}_{\boldsymbol{X}} \left[ \sum_i (f_i(\boldsymbol{A}, \boldsymbol{X}) - \boldsymbol{F}_i)(\boldsymbol{X}_i - \boldsymbol{F}_i)|\boldsymbol{A}, \boldsymbol{F} \right] \tag{16}$$

$$= \sum_i \mathbb{E}_{\boldsymbol{A}, \boldsymbol{F}} \left[ \mathbb{E}\big[(f_i(\boldsymbol{A}, \boldsymbol{X}) - \boldsymbol{F}_i)(\boldsymbol{X}_i - \boldsymbol{F}_i)|\boldsymbol{A}, \boldsymbol{F}\big] - \right.$$

$$\left. \mathbb{E}\big[f_i(\boldsymbol{A}, \boldsymbol{X}) - \boldsymbol{F}_i|\boldsymbol{F}\big] \mathbb{E}\big[\boldsymbol{X}_i - \boldsymbol{F}_i|\boldsymbol{A}, \boldsymbol{F}\big] \right] \tag{17}$$

$$= \sum_i \mathbb{E}_{\boldsymbol{A}, \boldsymbol{F}} \left[ \mathrm{Cov}(f_i(\boldsymbol{A}, \boldsymbol{X}) - \boldsymbol{F}_i, \boldsymbol{X}_i - \boldsymbol{F}_i|\boldsymbol{A}, \boldsymbol{F}) \right] \tag{18}$$

$$= \sum_i \mathbb{E}_{\boldsymbol{A}, \boldsymbol{F}} \left[ \mathrm{Cov}(f_i(\boldsymbol{A}, \boldsymbol{X}), \boldsymbol{X}_i|\boldsymbol{A}, \boldsymbol{F}) \right], \tag{19}$$

where $i$ sums over all dimensions $\{0, \cdots, d|V| - 1\}$, $f_i$ and $\boldsymbol{X}_i$ denotes the $i$-th element of the flattened matrices. Note that we employ $\mathbb{E}[\boldsymbol{X} - \boldsymbol{F}|\boldsymbol{A}, \boldsymbol{F}] = \boldsymbol{0}$ to let Equation (17) hold, according to the definition of latent graphs. Letting $J$ be a uniformly sampled subset of all node indices $\{0, \cdots, |V| - 1\}$, the right hand side of the above equation satisfies

$$\mathrm{RHS} = \mathbb{E}_J \frac{|V|}{|J|} \sum_{j \in J} \sum_{k=0}^{d-1} \mathbb{E}_{\boldsymbol{A}, \boldsymbol{F}} \left[ \mathrm{Cov}(f_{jd+k}(\boldsymbol{A}, \boldsymbol{X}), \boldsymbol{X}_{jd+k}|\boldsymbol{A}, \boldsymbol{F}) \right], \tag{20}$$

where $f_{jd+k} \in \mathbb{R}$ and $X_{jd+k} \in \mathbb{R}$ denote the $(jd + k)$-th element of corresponding matrices, *i.e.*, the $k$-th element of the node $v_j$, whereas $X_J \in \mathbb{R}^{|J| \times d}$ denotes the feature matrix of nodes in $V_J$. Given the bounded variance $\mathrm{Var}(\boldsymbol{X}_i) \leq \sigma^2, \forall i$, we bound the above term as

$$\mathrm{RHS} = \mathbb{E}_J \frac{|V|}{|J|} \sum_{j \in J, k} \mathbb{E}_{\boldsymbol{A}, \boldsymbol{F}} \left[ \mathrm{Cov}(f_{jd+k}(\boldsymbol{A}, \boldsymbol{X}) - f_{jd+k}(\boldsymbol{A}, \boldsymbol{X}_{J^c}), \boldsymbol{X}_{jd+k}|\boldsymbol{A}, \boldsymbol{F}) \right] \tag{21}$$

$$\leq \mathbb{E}_J \frac{|V|}{|J|} \sum_{j \in J, k} \mathbb{E}_{\boldsymbol{A}, \boldsymbol{F}} \left[ \mathrm{Var}(f_{jd+k}(\boldsymbol{A}, \boldsymbol{X}) - f_{jd+k}(\boldsymbol{A}, \boldsymbol{X}_{J^c})|\boldsymbol{A}, \boldsymbol{F}) \cdot \mathrm{Var}(\boldsymbol{X}_{jd+k}) \right]^{1/2} \tag{22}$$

$$\leq |V| \mathbb{E}_J \left( \frac{1}{|J|} \sum_{j \in J, k} \mathbb{E}_{\boldsymbol{A}, \boldsymbol{F}} \left[ \mathrm{Var}(f_{jd+k}(\boldsymbol{A}, \boldsymbol{X}) - f_{jd+k}(\boldsymbol{A}, \boldsymbol{X}_{J^c})|\boldsymbol{A}, \boldsymbol{F}) \cdot \sigma^2 \right] \right)^{1/2} \tag{23}$$

$$\leq \sigma|V| \mathbb{E}_J \left( \frac{1}{|J|} \sum_{j \in J, k} \mathbb{E}_{\boldsymbol{A}, \boldsymbol{F}} \left[ \mathbb{E}\big[ [f_{jd+k}(\boldsymbol{A}, \boldsymbol{X}) - f_{jd+k}(\boldsymbol{A}, \boldsymbol{X}_{J^c})]^2|\boldsymbol{A}, \boldsymbol{F} \big] \right] \right)^{1/2} \tag{24}$$

$$= \sigma|V| \mathbb{E}_J \left( \frac{1}{|J|} \sum_{j \in J, k} \mathbb{E}\big[ f_{jd+k}(\boldsymbol{A}, \boldsymbol{X}) - f_{jd+k}(\boldsymbol{A}, \boldsymbol{X}_{J^c}) \big]^2 \right)^{1/2} \tag{25}$$

$$= \sigma |V| \mathbb{E}_J \left( \frac{1}{|J|} \mathbb{E} \, \| f_J(\boldsymbol{A}, \boldsymbol{X}) - f_J(\boldsymbol{A}, \boldsymbol{X}_{J^c}) \|^2 \right)^{1/2}. \tag{26}$$

Above inequalities and equations are derived based on the fact that $f_J(\boldsymbol{A}, \boldsymbol{X}_{J^c})$ does not correlate to $\boldsymbol{X}_{jd+k}$ as $j \notin J^c$ for Equation (21), the Cauchy-Schwarz inequality for Inequality (22), and $(\mathbb{E} X)^2 \leq \mathbb{E} X^2$ for Inequality (23). We complete the proof of Theorem 1 by combining Equation (15) and Inequality (26),

$$\mathbb{E} \left[ \| f(\boldsymbol{A}, \boldsymbol{X}) - \boldsymbol{F} \|^2 + \| \boldsymbol{X} - \boldsymbol{F} \|^2 \right] = \mathbb{E} \, \| f(\boldsymbol{A}, \boldsymbol{X}) - \boldsymbol{X} \|^2 + 2 \langle f(\boldsymbol{A}, \boldsymbol{X}) - \boldsymbol{F}, \boldsymbol{X} - \boldsymbol{F} \rangle \tag{27}$$

$$\leq \mathbb{E} \, \| f(\boldsymbol{A}, \boldsymbol{X}) - \boldsymbol{X} \|^2 + 2 \sigma |V| \mathbb{E}_J \left( \frac{1}{|J|} \mathbb{E} \, \| f_J(\boldsymbol{A}, \boldsymbol{X}) - f_J(\boldsymbol{A}, \boldsymbol{X}_{J^c}) \|^2 \right)^{1/2}. \tag{28}$$

$$\square$$

## B  PROOF OF COROLLARY 1 AND 2

*Proof.* We first prove Corollary 1. Consider an $\ell$-Lipschitz continuous prediction head with respect to $l_2$-norm consists of fully connected layers. We have

$$\| f_J(\boldsymbol{A}, \boldsymbol{X}) - f_J(\boldsymbol{A}, \boldsymbol{X}_{J^c}) \|_2 = \| \mathcal{D}(\boldsymbol{H}_J) - \mathcal{D}(\boldsymbol{H}'_J) \|_2 \leq \ell \, \| \boldsymbol{H}_J - \boldsymbol{H}'_J \|_2. \tag{29}$$

We therefore have the following inequality

$$\mathbb{E} \, \| f_J(\boldsymbol{A}, \boldsymbol{X}) - f_J(\boldsymbol{A}, \boldsymbol{X}_{J^c}) \|_2^2 \leq \mathbb{E} \left[ \ell^2 \, \| \boldsymbol{H}_J - \boldsymbol{H}'_J \|_2^2 \right]. \tag{30}$$

We apply the above inequality to Theorem 1 and obtain the following inequality

$$\mathbb{E} \left[ \| f(\boldsymbol{A}, \boldsymbol{X}) - \boldsymbol{F} \|^2 + \| \boldsymbol{X} - \boldsymbol{F} \|^2 \right]$$

$$\leq \mathbb{E} \, \| f(\boldsymbol{A}, \boldsymbol{X}) - \boldsymbol{X} \|^2 + 2 \sigma |V| \mathbb{E}_J \left( \frac{1}{|J|} \mathbb{E} \, \| f_J(\boldsymbol{A}, \boldsymbol{X}) - f_J(\boldsymbol{A}, \boldsymbol{X}_{J^c}) \|^2 \right)^{1/2} \tag{31}$$

$$\leq \mathbb{E} \, \| f(\boldsymbol{A}, \boldsymbol{X}) - \boldsymbol{X} \|^2 + 2 \sigma |V| \mathbb{E}_J \left( \frac{1}{|J|} \mathbb{E} \left[ \ell^2 \, \| \boldsymbol{H}_J - \boldsymbol{H}'_J \|_2^2 \right] \right)^{1/2} \tag{32}$$

$$= \mathbb{E} \, \| f(\boldsymbol{A}, \boldsymbol{X}) - \boldsymbol{X} \|^2 + 2 \sigma |V| \ell \mathbb{E}_J \left( \mathbb{E} \, \| \boldsymbol{H}_J - \boldsymbol{H}'_J \|_2^2 / |J| \right)^{1/2}, \tag{33}$$

which completes the proof of Corollay 1.

Similarly, for Corollay 2, we have

$$\| f_J(\boldsymbol{A}, \boldsymbol{X}) - f_J(\boldsymbol{A}, \boldsymbol{X}_{J^c}) \|_2 = \| \mathcal{D}(\boldsymbol{H}_J) - \mathcal{D}(\boldsymbol{H}'_J) \|_2 \leq \ell \, \| \boldsymbol{H}_J - \boldsymbol{H}'_J \|_2. \tag{34}$$

Given an $\ell_r$-Bilipschitz continuous readout function $\mathcal{R}$, the following inequalities hold,

$$\frac{1}{\ell_r} \| \boldsymbol{H}_J - \boldsymbol{H}'_J \|_2 \leq \| \mathcal{R}(\boldsymbol{H}_J) - \mathcal{R}(\boldsymbol{H}'_J) \|_2 \leq \ell_r \, \| \boldsymbol{H}_J - \boldsymbol{H}'_J \|_2. \tag{35}$$

We therefore have

$$\mathbb{E} \left[ \| f(\boldsymbol{A}, \boldsymbol{X}) - \boldsymbol{F} \|^2 + \| \boldsymbol{X} - \boldsymbol{F} \|^2 \right]$$

$$\leq \mathbb{E} \, \| f(\boldsymbol{A}, \boldsymbol{X}) - \boldsymbol{X} \|^2 + 2 \sigma |V| \ell \mathbb{E}_J \left( \mathbb{E} \, \| \boldsymbol{H}_J - \boldsymbol{H}'_J \|_2^2 / |J| \right)^{1/2} \tag{36}$$

$$\leq \mathbb{E} \, \| f(\boldsymbol{A}, \boldsymbol{X}) - \boldsymbol{X} \|^2 + 2 \sigma |V| \ell \ell_r \mathbb{E}_J \left( \mathbb{E} \, \| \mathcal{R}(\boldsymbol{H}_J) - \mathcal{R}(\boldsymbol{H}'_J) \|_2^2 / |J| \right)^{1/2} \tag{37}$$

$$= \mathbb{E} \, \| f(\boldsymbol{A}, \boldsymbol{X}) - \boldsymbol{X} \|^2 + 2 \sigma |V| k \ell \mathbb{E}_J \left( \mathbb{E} \, \| \boldsymbol{z} - \boldsymbol{z}' \|_2^2 / |J| \right)^{1/2}, \tag{38}$$

which completes the proof of Corollay 2.

$$\square$$

## C  Pseudo-code for Computing LaGraph Objectives

The pseudo-code for computing LaGraph objectives for node-level representations and graph-level representations are shown in Algorithms 1 and 2.

---

**Algorithm 1** LaGraph objective: node-level

---

**Inputs:** A mini-batch of graphs $\{G_1, \cdots, G_N\}$, the encoder $\mathcal{E}$, the prediction head $\mathcal{D}$, and the hyper-parameter $\alpha$. $\qquad \triangleright G_i = (\boldsymbol{A}_i, \boldsymbol{X}_i)$

$\quad$ **for** $i$ in $1, \cdots, N$ **do**

$\quad\quad$ Generate random $J_i \in \{0,1\}^{|V_i| \times 1}, \boldsymbol{M} \in \mathbb{R}^{|V_i| \times d}$

$\quad\quad \boldsymbol{X}_{i, J_i^c} \leftarrow \mathbb{1}_{J_i^c} \odot \boldsymbol{X} + \mathbb{1}_{J_i} \odot \boldsymbol{M}$ $\qquad\qquad\qquad$ $\triangleright$ Randomly mask nodes

$\quad\quad \boldsymbol{H}_i \leftarrow \mathcal{E}(\boldsymbol{A}_i, \boldsymbol{X}_i)$ $\qquad\qquad\qquad\qquad$ $\triangleright$ Compute node representations

$\quad\quad \boldsymbol{H}_i' \leftarrow \mathcal{E}(\boldsymbol{A}_i, \boldsymbol{X}_{i, J_i^c})$

$\quad\quad \boldsymbol{X}_{rec} \leftarrow \mathcal{D}(\boldsymbol{A}_i, \boldsymbol{H}_i)$ $\qquad\qquad\qquad\qquad$ $\triangleright$ Reconstructed node attributes

$\quad\quad \ell_{rec,i} \leftarrow \|\boldsymbol{X}_{rec,i} - \boldsymbol{X}_i\|^2 / |V_i|$

$\quad\quad \ell_{inv,i} \leftarrow \|\mathbb{1}_{J_i} \odot \boldsymbol{H}_i - \mathbb{1}_{J_i} \odot \boldsymbol{H}_i'\|^2$

$\quad$ **end for**

$\quad L(\mathcal{E}, \mathcal{D}; \{G_1, \cdots, G_N\}) = \frac{1}{N} \sum_i \ell_{rec,i} + \alpha (\sum_i \ell_{inv,i} / \sum_i |J_i|)^{1/2}$

---

**Algorithm 2** LaGraph objective: graph-level

---

**Inputs:** A mini-batch of graphs $\{G_1, \cdots, G_N\}$, the encoder $\mathcal{E}$, the prediction head $\mathcal{D}$, the readout function $\mathcal{R}$, and the hyper-parameter $\alpha$. $\qquad \triangleright G_i = (\boldsymbol{A}_i, \boldsymbol{X}_i)$

$\quad$ **for** $i$ in $1, \cdots, N$ **do**

$\quad\quad$ Generate random $J_i \in \{0,1\}^{|V_i| \times 1}, \boldsymbol{M} \in \mathbb{R}^{|V_i| \times d}$

$\quad\quad \boldsymbol{X}_{i, J_i^c} \leftarrow \mathbb{1}_{J_i^c} \odot \boldsymbol{X} + \mathbb{1}_{J_i} \odot \boldsymbol{M}$ $\qquad\qquad\qquad$ $\triangleright$ Randomly mask nodes

$\quad\quad \boldsymbol{H}_i \leftarrow \mathcal{E}(\boldsymbol{A}_i, \boldsymbol{X}_i)$ $\qquad\qquad\qquad\qquad$ $\triangleright$ Compute node embeddings

$\quad\quad \boldsymbol{H}_i' \leftarrow \mathcal{E}(\boldsymbol{A}_i, \boldsymbol{X}_{i, J_i^c})$

$\quad\quad \boldsymbol{z}_i \leftarrow \mathcal{R}(\boldsymbol{H}_i)$ $\qquad\qquad\qquad\qquad\qquad$ $\triangleright$ Readout graph representations

$\quad\quad \boldsymbol{z}_i' \leftarrow \mathcal{R}(\boldsymbol{H}_i')$

$\quad\quad \boldsymbol{X}_{rec} \leftarrow \mathcal{D}(\boldsymbol{A}_i, \boldsymbol{H}_i)$ $\qquad\qquad\qquad\qquad$ $\triangleright$ Reconstructed node attributes

$\quad\quad \ell_{rec,i} \leftarrow \|\boldsymbol{X}_{rec,i} - \boldsymbol{X}_i\|^2 / |V_i|$

$\quad\quad \ell_{inv,i} \leftarrow \|\boldsymbol{z}_i - \boldsymbol{z}_i'\|^2$

$\quad$ **end for**

$\quad L(\mathcal{E}, \mathcal{D}; \{G_1, \cdots, G_N\}) = \frac{1}{N} \sum_i \ell_{rec,i} + \alpha (\sum_i \ell_{inv,i} / \sum_i |J_i|)^{1/2}$

---

## D  Discussion on the differences to BGRL and Barlow-Twin objectives

In spite of the similarity in the objective of BGRL, Barlow-Twin, the consistency regularization (Wei et al., 2020), and the invariance term of the LaGraph objective, we argue that there are substantial differences between the invariance term in our objective and the BGRL and Barlow-Twins objectives. Regarding how the objective is computed, the invariance term is only computed for the masked nodes in Theorem 1 and objective for node-level representations, in contrast to the BGRL and Barlow-Twins objectives where all nodes are computed. It is worth noting that the objective is an upper bound to the latent graph prediction only if the invariance is computed on the masked nodes, according to the derivation during Proof of Theorem 1. Intuitively, during the computation of a node representation, the invariance term enforces the encode to capture less information from the node itself and more contextual information. If we also include unmasked nodes like BGRL, the invariance will lead to a contradicted effect, i.e., enforcing encoders to capture less information from contextual nodes, as it lets the representation remain consistent when its neighbor nodes (who are masked) are changed. We believe the derivation and the intuition of the proposed objective can provide guidance on how to adopt the consistent regularization into SSL studies.

# E    EXPERIMENT SETTINGS AND MODEL CONFIGURATIONS

**Dataset Statistics.**    Statistics including number of graphs, averaged number of nodes, averaged number of edges, and node attribute dimensions are summarized in Table 4.

Table 4: Summary and statistics of common graph datasets for self-supervised learning.

| Datasets | Evaluation task | # graphs | Avg. nodes | Avg. edges | # features |
|---|---|---|---|---|---|
| **NCI1** | | 4110 | 29.87 | 32.30 | 37 |
| **PROTEINS** | | 1178 | 39.06 | 72.82 | 3 |
| **DD** | | 188 | 284.32 | 715.66 | 89 |
| **MUTAG** | Graph-level | 1113 | 17.93 | 19.79 | 7 |
| **COLLAB** | classification | 5000 | 74.49 | 2457.78 | 1 |
| **RDT-B** | | 2000 | 429.63 | 497.75 | 1 |
| **RDT-M5K** | | 4999 | 508.52 | 594.87 | 1 |
| **IMDB-B** | | 1000 | 19.77 | 96.53 | 1 |
| **Amazon Computer** | Transductive | 1 | 13,752 | 245,861 | 767 |
| **Amazon Photos** | Node-level | 1 | 7,650 | 119,081 | 745 |
| **Coauthor CS** | classification | 1 | 81,894 | 81,894 | 6,806 |
| **Coauthor Physics** | | 1 | 247,962 | 247,962 | 8,415 |
| **PPI** | Inductive | 24 | 2,373 | 34,133 | 50 |
| **Flickr** | Node-level | 1 | 89,250 | 899,756 | 500 |
| **Reddit** | classification | 1 | 232,965 | 11,606,919 | 602 |

**Transductive and Inductive Settings.**    For node-level tasks, data typically consists of one or more large graphs with nodes and edges. The goal of self-supervised learning on such node-level datasets is to learn high-quality node representation that facilitates downstream node classification tasks. Our node-level experiments are divided into two categories, namely, transductive and inductive tasks. **Transductive self-supervised learning** of node representation allows utilization of all data at hand to pre-train GNNs for downstream tasks. Although labels of nodes are not visible during pre-training, patterns and information present in all nodes are observed. In contrast to transductive learning, **inductive self-supervised learning** only allows using a portion of data to pre-train GNNs, while holding out a certain amount of data for downstream tasks. Our inductive tasks include two cases. First, the PPI dataset consists of 24 graphs, and the training and testing nodes are split by graphs. In this case, the inductive task is considered across multiple graphs. In other words, node representations are learned from training graphs, and the encoder is evaluated on testing graphs. Second, Flickr and Reddit each consist of only one graph, the training and testing nodes are from the same graph. During self-supervised training, all test nodes are masked-out. During evaluation, all training nodes are masked-out, i.e., test nodes are unseen nodes of the graph. For both cases of inductive learning, data used during the self-supervised training stage and data used during evaluation stage are distinct, but the feature dimensionality should be the same for data used in both stages.

**Experimental Details.**    We train graph-level datasets on a single 11GB GeForce RTX 2080 Ti GPU, and train node-level datasets on a single 56GB Nvidia RTX A6000 GPU. Our experiments are implemented with PyTorch 1.7.0 and PyTorch Geometric 1.7.0. All neural networks employ batch normalization (Ioffe and Szegedy, 2015), and are optimized with Adam optimizer. We initialize GNNs with Xavier initialization (Glorot and Bengio, 2010).

**Models for Graph-Level Datasets.**    We employ a 3-layer GIN (Xu et al., 2019) as the graph encoder $\mathcal{E}$, and a 2-layer MLP as the decoder $\mathcal{D}$. Following GraphCL (You et al., 2020), we use a hidden dimension of size 32 and concatenate the embedding at each encoding layer to obtain the final representation. To fulfill the conditions in Corollary 2, we apply global sum pooling as the readout function $\mathcal{R}$. The obtained graph representation is then taken by a SVM classifier with a 10-fold evaluation. For graph datasets that do not come with node attributes, we apply the one-hot

vector of the degree for each node as the node attributes so that the node degrees are reconstructed. Certain thresholds for max degrees are applied to reduce computational cost and avoid over sparse node features. The neural network is trained using the loss described in Equation (6). We mask all attributes of the sampled nodes with Gaussian noise. Detailed training configurations including mask ratio, standard deviation of noise, weight scalar $\alpha'$ and threshold for max degrees are shown in Table 5. Note that we do not include carefully designed implementation mechanisms by BGRL, such as stop gradients, EMA, and batch normalization at the last layer.

**Models for Node-Level Datasets.** For node-level datasets, we employ a 2-layer GCN (Kipf and Welling, 2017) as the graph encoder $\mathcal{E}$, and a linear layer or an MLP as the decoder $\mathcal{D}$. We use a hidden dimension of size 512 at each encoding layer. The neural network is trained using the loss described in Equation (3). We uniformly employ the weight scalar $\alpha'$ of 2 as we observed that the model performance is not sensitive to the selection of $\alpha'$ within the range $[1, 100]$. We obtain the final node representation by concatenating the original feature with the embedding from the last layer of encoder. The intuition of this is based on the Bayesian rule where the learned encoder provides the prior knowledge (Ulyanov et al., 2018) of data distribution whereas the given graph data serves as the observed samples. And the posteriori should be based on a combination of the priori (encoder output) and the observed data itself (Laine et al., 2019). Node representation is then taken by a logistic regression classifier that is trained using the cross entropy (CE) loss with a learning rate of $0.01$. Detailed training configurations including mask ratio, standard deviation of noise, number of encoder and decoder layers, learning rate and weight decay of the graph neural network, training epochs and weight decay of the logistic regression classifier are shown in Table 6. To split train, valid and test sets, we use the public split used in (Shchur et al., 2018) for Coauthor and Amazon, (Zitnik and Leskovec, 2017; Hamilton et al., 2017; Zeng et al., 2019) for PPI, Reddit and Flickr provided by PyTorch Geometric. Note that we do not include implementation mechanisms by BGRL, such as stop gradients, EMA, and batch normalization at the last layer.

Table 5: Model configurations for graph-level datasets.

|  | NCI1 | PROTEINS | DD | MUTAG | COLLAB | RDT-B | RDT-M5K | IMDB-B |
|---|---|---|---|---|---|---|---|---|
| Mask ratio | 0.05 | 0.3 | 0.1 | 0.05 | 0.05 | 0.05 | 0.05 | 0.05 |
| Noise SD | 0.5 | 2 | 0.5 | 0.5 | 0.5 | 0.5 | 0.5 | 0.5 |
| Weight scalar $\alpha'$ | 10 | 1 | 10 | 10 | 10 | 10 | 10 | 10 |
| Degree threshold | – | – | – | – | 128 | – | – | 64 |
| Learning rate | $10^{-5}$ | $10^{-5}$ | $10^{-5}$ | $10^{-5}$ | $10^{-4}$ | $10^{-3}$ | $10^{-4}$ | $10^{-4}$ |

Table 6: Model configurartions for node-level datasets.

|  | Am.Computers | Am.Photos | CoautherCS | CoauthorPhy | PPI | Flickr | Reddit |
|---|---|---|---|---|---|---|---|
| Mask ratio | 0.05 | 0.05 | 0.05 | 0.05 | 0.05 | 0.01 | 0.05 |
| Noise SD | 0.5 | 0.5 | 0.005 | 0.5 | 0.5 | 0.5 | 0.5 |
| Decoder layers | 1 | 1 | 1 | 2 | 2 | 2 | 2 |
| Learning rate | $10^{-4}$ | $10^{-5}$ | $10^{-3}$ | $10^{-3}$ | $10^{-3}$ | $10^{-4}$ | $10^{-3}$ |
| Weight decay | 0 | $10^{-4}$ | 0 | 0 | $10^{-5}$ | 0 | 0 |
| LogReg epochs | 400 | 400 | 400 | 300 | 200 | 200 | 500 |
| LogReg WD | $10^{-3}$ | $10^{-3}$ | $10^{-3}$ | $10^{-3}$ | 0 | $10^{-3}$ | $10^{-3}$ |

## F    EXPERIMENTAL RESULTS UNDER SEMI-SUPERVISED SETTING

For graph-level datasets, we perform semi-supervised experiments with 10% label rate using both GIN and GCN. All experiments are conducted with the same random seed to avoid randomness in data split and initialization. Under the setting of random initialization followed by supervised learning, the GNN is random initialized without pre-training. Under the setting of LaGraph followed by supervised learning, the GNN is pre-trained with the proposed LaGraph framework. Weights of GNNs are fine-tuned during the supervised learning with 10% labels. For each dataset, learning rate and epoch number for pre-training are the same as what we use under unsupervised setting. For fine-tuning, learning rate is selected from $\{10^{-3}, 10^{-4}, 10^{-5}\}$, and epoch number is selected

from $\{5, 10, 15, 20\}$. The results shown in Table 7 and Table 8 indicate that our proposed LaGraph framework is also effective for semi-supervised learning with different GNN backbones.

Table 7: GIN results for Semi-supervised learning.

| | NCI1 | PROTEINS | DD | MUTAG | COLLAB | RDT-B | RDT-M5K | IMDB-B |
|---|---|---|---|---|---|---|---|---|
| Rand. Init. + 10% supervised | 76.67 | 75.29 | 76.66 | 86.67 | 77.54 | 85.45 | 56.03 | 72.70 |
| LaGraph + 10% supervised | 80.19 | 76.10 | 77.93 | 91.40 | 78.04 | 89.65 | 56.43 | 74.30 |

Table 8: GCN results for Semi-supervised learning.

| | NCI1 | PROTEINS | DD | MUTAG | COLLAB | RDT-B | RDT-M5K | IMDB-B |
|---|---|---|---|---|---|---|---|---|
| Rand. Init. + 10% supervised | 75.47 | 74.48 | 77.33 | 84.59 | 79.02 | 85.30 | 53.67 | 72.90 |
| LaGraph + 10% supervised | 78.18 | 76.28 | 78.86 | 85.12 | 80.12 | 90.35 | 55.33 | 75.10 |

## G  ADDITIONAL ABLATION STUDIES.

**Ablation on Optimizing Different Objectives.** We empirically compare the effect of different upper bounds on graph-level datasets. In addition to objectives described in Corollary 2, we further train the graph encoder with the upper bound described in Theorem 1, who applies invariance regularization on the reconstructed node features. In addition, as node attributes in many graph-level datasets are formed as one-hot vectors of node type, we also provide the results of using two corresponding multinomial versions of the objective. In particular, we replace the reconstruction term by the cross-entropy between $f(A, X)$ and $X$ and, if computed on the outputs, the invariance term by the KL-divergence between $f_J(A, X)$ and $f_J(A, X_{J^c})$. Note that the multinomial versions are no longer strictly upper bounds of supervised latent graph prediction. In Table 10, we show the results obtained under four objectives above, namely, to compute invariance on on-embedding (MSE-Embed), on-output (MSE-Output), and their corresponding multinomial versions (CE-Embed and CE-Output), respectively. Results indicate that there is no significant difference among the four versions on most datasets, while MSE-Embed and CE-Embed generally tend to be more stable and achieve higher performance on MUTAG, RDT-B, and RDT-M5K.

Table 9: Effect of training with different objectives on graph-level datasets.

| | NCI1 | PROTEINS | DD | MUTAG | COLLAB | RDT-B | RDT-M5K | IMDB-B |
|---|---|---|---|---|---|---|---|---|
| MSE-Embed | **79.9±0.5** | 75.2±0.3 | **78.1±0.3** | **90.2±1.3** | 77.6±0.1 | 90.4±0.9 | **56.4±0.2** | **73.7±0.7** |
| MSE-Output | 79.9±0.7 | 75.0±0.4 | 78.1±0.8 | 89.2±2.1 | **77.7±0.1** | 89.8±0.9 | 56.0±0.4 | 73.4±0.6 |
| CE-Embed | 79.9±0.5 | **75.2±0.3** | 78.1±0.4 | 90.1±1.0 | 77.6±0.2 | **90.5±1.3** | 56.3±0.4 | 73.7±0.7 |
| CE-Output | 79.9±0.7 | 75.2±0.4 | 78.1±0.4 | 89.3±2.7 | 77.6±0.1 | 89.4±1.8 | 55.7±0.2 | 73.5±0.5 |

**Ablation on Concatenated Representations for Node-Level Datasets.** For the node-level datasets, We obtain the final node representation by concatenating the original feature with the embedding from the last layer of encoder, due to the intuition discussed in Appendix . Results in Table 10 compare the performance of representations with or without concatenations. The removal of the concatenation leads to reduced performance on four of the seven datasets, and performance gain on the rest datasets including the most challenging PPI. The results indicates that the concatenation generally positively contributes to the final performance. However, the conclusion still holds that, on node-level datasets, LaGraph can provide significant performance gain on challenging datasets where there is a gap between SSL and supervised performance. Meanwhile, the performance of LaGraph is on par with the performance of supervised learning and the SOTA method BGRL on datasets that are less challenging.

Table 10: Effect of performing concatenation with node features.

| Dataset | Am. Comp. | Am. Pht. | Co. CS | Co. Phy | PPI | Flickr | Reddit |
|---|---|---|---|---|---|---|---|
| With concat | 88.0±0.3 | **93.5±0.4** | **93.3±0.2** | **95.8±0.1** | 74.6±0.0 | 51.3±0.1 | **95.2±0.0** |
| W/o concat | **88.8±0.3** | 92.7±0.4 | 92.6±0.2 | 95.3±0.1 | **75.2±0.0** | **51.6±0.1** | 94.8±0.0 |

## H  ADDITIONAL DISCUSSION ON POTENTIAL LIMITATIONS AND FUTURE DIRECTIONS

In this section, we discuss several limitations of the proposed method and their solutions or related future directions.

**Non-significant improvement compared to BGRL on transductive tasks.**  From the experimental perspective, we admit that BGRL is a quite strong baseline method for transductive tasks. As the results are already on the same level as the performance of supervised training, it is very difficult to further obtain significant improvements. However, when it comes to inductive tasks, where there is still a significant gap between the performance of BGRL and supervised learning, our method is able to bring significant improvements in performance. Therefore, we argue that the non-significant performance boost on some transductive datasets does not degrade our main conclusion about the effectiveness of our method.

**Unattributed graphs.**  Although, in this work, we mainly focus graphs with attributed nodes, there exists cases where the nodes are unattributed and its all information are contained in the graph topology, especially for some graph-level datasets. In such cases, we follow a common solution to consider the one-hot vectors of node degrees as the node attributes and our objective performs reconstruction on the node degree. To avoid inconsistency between training and testing graphs in their range of degrees, we introduce thresholds to the node degrees, *i.e.*, the degree of a node is considered as $k$ if it exceeds $k$. The current solution can capture topological information of a graph at some degree. However, there can be better solutions capturing full topological information of graphs. A potential direction is to perform connectivity reconstruction with invariance of representations to changing in the input edge set. Although the described approach does not currently fit into our theoretical framework, it is possible to derive similar objectives (e.g., upper bound to link prediction objective) following a similar idea.

**Scaling-up issue.**  The scaling-up of graph neural networks becomes an emerging topic. Many existing self-supervised methods may suffer from the scaling-up issue, when the graph scales up to billions of nodes and edges. Although we do not perform experimental study on extremely large graphs, we perform ablation studies to demonstrate the robustness of our method to the training schemes of sampling subgraphs (mini-batches of nodes) for each training iterations.

**Performance of SSL methods on unsupervised downstream tasks.**  Performing linear evaluation with supervised downstream tasks on learned representations is the most common way to evaluate the performance of SSL methods. However, evaluation performance on graph-specific unsupervised tasks such as overlapping community detection (Xie et al., 2013) are seldom studied. Further investigations in unsupervised downstream task are required to fully demonstrate the effectiveness of self-supervised learning methods.

