# OpenReview forum: "Self-Supervised Representation Learning via Latent Graph Prediction"
_ICLR.cc/2022/Conference — ICLR 2022 Submitted_

### Official Review · Reviewer_e11u · 2021-10-20

**Correctness:** 4
**Technical Novelty And Significance:** 3
**Empirical Novelty And Significance:** 2
**Recommendation:** 5
**Confidence:** 3

**Main Review:**

Strengths
--

* The paper studied self-supervised learning for GNNs that is less explored than other domains.
* The authors provide theoretical analyses of the proposed method. The relation between the proposed method and previous work is discussed with theoretical analysis.
* The memory consumption is much less than a strong baseline GraphCL and works well (e.g., relatively small degradation) with a relatively small mini-batch size.

Weaknesses
--
* The proposed method and losses are a simple variation of existing methods. Although the authors present their works as latent graph prediction, it is indeed a simple consistency regularization with noise on randomly sampled nodes. Since the method is proposed as an unsupervised method, the authors impose the consistency regularization on hidden representations, which are the output of encoder (GNNs) rather than predictions/softmax values.
* Although the relation between latent graph prediction and consistency regularization is interesting, the relation and analyses barely provide intuitions to improve the consistency loss.
* It is not clear whether the performance gain is statistically significant.

Questions
--
* Do you use stop gradient along the branch for original graphs?
* Is the proposed method applicable to link prediction?
* Why do you focus on only masked nodes for the consistency of hidden representations in (5) and (6)?
* Is the substitution of node features with random noise for randomly sampled nodes more effective than dropNode, which are used in previous work such as GRAND [1]?

[1] Feng, Wenzheng, et al. "Graph Random Neural Network for Semi-Supervised Learning on Graphs." NeurIPS, 2020.

**Summary Of The Paper:**

The authors propose a self-supervised learning (SSL) framework for graph neural networks by predicting an unobserved latent graph. The proposed self-supervised learning loss is derived from upper bounds to the original objective functions for predicting latent graphs. The proposed method is robust to small sample size and small batch size.

**Summary Of The Review:**

This paper studies a self-supervised learning framework to perform representation learning for graph neural networks. The authors provide theoretical discussion about latent graph prediction and related domains. Even though many theoretical analyses and derivations are provided, the resulting algorithm is too close to existing methods, and the latent graph prediction perspective makes it more difficult to understand the proposed method. It has strengths with interesting theoretical analyses, but the impact and the novelty of the resulting algorithms are limited.

---

> ### Author Response · Authors · 2021-11-11
> **Response to comments by Reviewer e11u (con'd)**
>
> > Why do you focus on only masked nodes for the consistency of hidden representations in (5) and (6)?
>
> It is based on the derivation in the proof of theorem 1. In particular, the current invariance term is an upper bound to the term <f(\bm{A},\bm X)-\bm F, \bm X - \bm F> in Equation 15. When including all nodes for consistency computation, the objective is no longer an upper bound of latent graph prediction. As mentioned in the previous response, intuitively, during the computation of a node representation, the invariance term enforces the encode to capture less information from the node itself and more contextual information. If we also include unmasked nodes, the invariance will lead to a contradicted effect, i.e., enforcing encoders to capture less information from contextual nodes, as it lets the representation remain consistent when its neighbor node (who is masked) is changed.
>
> We thank the reviewer for pointing out this important detail. It can be considered as a substantial difference between our objective and previous similar objectives based on different groundings. We have updated several statements in the draft to make it more clear.
>
>
> > Is the substitution of node features with random noise for randomly sampled nodes more effective than dropNode, which were used in previous work such as GRAND [1]?
>
> Under our theoretical framework, dropping a node may break the topology of a graph, which does not fit into our framework and should be considered as a different direction of inducing self-supervision. Indeed, the topological information can also provide good self-supervision to the learning of representations. We have included a discussion (unattributed graph) in Appendix~G to discuss several approaches that fit into our framework and future directions following the similar idea to fully adopt topology as self-supervision.

---

> ### Author Response · Authors · 2021-11-11
> **Response to comments by Reviewer e11u**
>
> We thank the reviewer for the valuable feedback and suggestions. Below are our item-wise responses to the comments
>
> > The proposed method and losses are a simple variation of existing methods. Although the authors present their works as latent graph prediction, it is indeed a simple consistency regularization with noise on randomly sampled nodes. Since the method is proposed as an unsupervised method, the authors impose the consistency regularization on hidden representations, which are the output of encoder (GNNs) rather than predictions/softmax values.
> > Although the relation between latent graph prediction and consistency regularization is interesting, the relation and analyses barely provide intuitions to improve the consistency loss.
>
> The invariance term in the proposed objective does share some similarities with the consistency regularization. However, we argue that the two objectives have **substantial differences** as they are based on different theoretical groundings and have different effects. As the reviewer mentioned in a later comment that “we focus on only masked nodes” when computing the invariance term, the two objectives are based on different motivations with different effects. Intuitively, during the computation of a node representation, the invariance term enforces the encode to capture less information from the node itself and more contextual information. If we also include unmasked nodes, the invariance will lead to a contradicted effect, i.e., enforcing encoders to capture less information from contextual nodes, as it lets the representation remain consistent when its neighbor nodes (who are masked) are changed. We believe the derivation and the intuition of the proposed objective can provide guidance on how to adopt the consistent regularization into SSL studies.
>
> To emphasize the above intuition and insights, we have included a discussion in Appendix.
>
>
> > It is not clear whether the performance gain is statistically significant.
>
> The computing of statistical significance requires a t-test based on sample-wise scores. Due to the time constraint, we are unable to reproduce all baseline methods for the sample-wise scores during the discussion session. We can visually see significant improvement on most graph-level datasets and the PPI dataset. Regarding the remaining node-level datasets, we admit that it is possible that the improvement can be statistically non-significant. However, for those datasets, existing baseline methods are already strong with a performance on the same level as supervised learning. It is very difficult to further provide a significant performance boost due to the saturation of existing performance. Taking all results into account, our conclusion on the effectiveness of the proposed method still holds. In particular, for datasets with saturated SSL performance, our method can achieve as good performance as existing SOTA methods. For datasets where a gap exists between the performance of SSL and supervised methods, our method is able to provide a significant performance improvement.
>
> We have updated our draft to include the above discussion in Appendix G.
>
>
> > Do you use stop gradients along the branch for original graphs?
>
> No. We do not use stop gradient nor additional carefully designed implementation details by BGRL such as EMA and batch-norm at the last layer. We have included a statement in the model configuration section in Appendix D to make this more clear.
>
> > Is the proposed method applicable to link prediction?
>
> Yes. The link prediction task still requires the computing of informative node representations (embeddings) and performing pairwise classification on top of the embeddings. As our method is capable of learning node-level representations, it can be applied to link prediction as to the downstream task.

---

> ### Author Response · Authors · 2021-11-23
> **Authors' follow-up on comments by Reviewer e11u**
>
> Thank you again for your valuable comments! We believe we have addressed your concerns in our previous responses. We hope that you could consider updating your score if we do have addressed your concerns. Also, please let us know if there are any additional concerns or feedback. Thank you!

---

> ### Author Response · Authors · 2021-11-28
> **Authors' follow-up**
>
> Dear Reviewer,
>
> Since the discussion period will end soon, could you kindly check our response and revision? We believe we have addressed all of your concerns and are looking forward to hearing from you.

---

### Official Review · Reviewer_X3u8 · 2021-11-02

**Correctness:** 3
**Technical Novelty And Significance:** 3
**Empirical Novelty And Significance:** 2
**Recommendation:** 6
**Confidence:** 3

**Main Review:**

However, the model assumption, that the conditional distribution of the observed graph is centered at the latent graph, is too strong, which is not natural at all. Meanwhile, it seems the introduced objective is just an upper bound, whose tightness should be further discussed.

Meanwhile, as claimed in Section 3, the introduced objective is highly correlated with the existing method, which is good for us the understand the method. However, the difference or superiority of the objective over the existing InforMax should also be highlighted to analyze the method deeper.

Although the motivation of eliminating the latent feature matrix F is impressive, the method adopted in this paper highly depends on the approximation of the upper bound, i.e., the number of subgraph N and hyperparameter alpha. This should be declared in the paper and the scenario that the proposed method is not suitable can also be discussed.

The notation in this paper is not professional, especially the use of subscript. $1_J, X_J, f_J, E_J, H_J; l_n, l_g, l_r$ It hinders me from fully understanding the meaning of some terms.

some minor concerns:

How is Eq.12 derived?

what does "Supervised" mean in Table 2?

**Summary Of The Paper:**

A method called LaGraph is proposed for semi-supervised graph representation learning. In particular, a new task named latent graph prediction is introduced, and the corresponding objective requiring no negativing samples is also derived. It is impressive that the proposed method requires no negative samples and works with a small batch size.

**Summary Of The Review:**

The idea of the proposed method is novel and interesting to me, but the assumption of latent graph about the generation process of graph is a bit strong.

---

> ### Author Response · Authors · 2021-11-10
> **Response to comments by Reviewer X3u8 (con'd)**
>
> > The notation in this paper is not professional.
>
> We thank the reviewer for pointing out the problem. For \ell_g, \ell_n, \ell_r, we have replaced them by \ell, \ell, and k to avoid unnecessary subscripts. For notations with subscripts J, we believe those subscripts are necessary. We have updated the draft to include more descriptions of the notations that may cause confusion. To be more clear, J is a subset of the node indices {0, · · ·, |V | − 1} and J^c is its complement set. In general, matrices with subscripts J or J^c denote that some rows (e.g., attributes of some nodes) are masked or dropped. 1_J is a vector with value 1 at indices in J, and 0 everywhere else.
>
> > How is Eq.12 derived?
>
> \arg\min_{\bm Z} \left[H(\bm Z)-H(\bm Z|\tilde{\bm X})\right] - \beta \left[H(\bm Z)-H(\bm Z|\bm X)\right]
>
> = \arg\min_{\bm Z} \beta H(\bm Z|\bm X) - (\beta-1) H(\bm Z),
>
> = \arg\min_{\bm Z} H(\bm Z|\bm X) - \lambda H(\bm Z),
>
> in which H(\bm Z|\tilde{\bm X})=0 because the encoder (compressor) is a deterministic function.
>
> > What does "Supervised" mean in Table 2?
>
> “Supervised” denotes the results by performing end-to-end supervised training on each dataset.

---

> ### Author Response · Authors · 2021-11-10
> **Response to comments by Reviewer X3u8**
>
> We thank the reviewer for the valuable feedback and suggestions. Below are our item-wise responses to the comments.
>
> > However, the model assumption, that the conditional distribution of the observed graph is centered at the latent graph, is too strong, which is not natural at all.
>
> Considering the case of noisy and clean data, such assumption is natural for many noise types such as Gaussian white noise and Poisson noise. We admit that the assumption is somehow strong. However, when we have zero knowledge about how an observed graph is generated from the latent graph, the “centered” assumption is the best (and most common) assumption we can make.
>
> > Meanwhile, it seems the introduced objective is just an upper bound, whose tightness should be further discussed.
>
> The proposed upper bound is not guaranteed to be the tightest. However, it is tighter when compared to other objectives (e.g., denoising autoencoder and BGRL objectives), detailed below.
>
> The tightness can be determined by inequalities 22-24 in the proof of Theorem 1, of which, the most important inequality scaling is “$Var(X_{jd+k}) \le \sigma^2$”. The bound becomes tighter when $\sigma$ is closer to $Var(X_{jd+k})$
>  til the equality holds. If we compare our objective with the denoising autoencoder and the BGRL objectives, we can find that the denoising autoencoder objective is a special case of ours when $\sigma=0$ (with only reconstruction term), whereas the BGRL objective is a special case when $\sigma\to\inf$ (with only invariance term). Both are looser than the proposed objective when considered as upper bounds.
>
>
> > Meanwhile, as claimed in Section 3, the introduced objective is highly correlated with the existing method, which is good for us to understand the method. However, the difference or superiority of the objective over the existing InfoMax should also be highlighted to analyze the method deeper.
>
> We discuss the relationship between our objective and the InfoMax objective which maximizes the mutual information between a local node representation and the graph representation. As mentioned in the discussion, the two results have similar effects, i.e., both encourage the node representation to include more contextual information. The major difference between the two objectives lies in their theoretical grounding. Regarding the effects, InfoMax allows GNN to capture longer-range (more hops) dependencies as a global view is used but may weaken the dependency of local contexts (as representations of all nodes are averaged for the global view). As they are different frameworks, it is hard to theoretically compare their superiority, in which case we can refer to the empirical comparisons.
>
> > Although the motivation of eliminating the latent feature matrix F is impressive, the method adopted in this paper highly depends on the approximation of the upper bound, i.e., the number of subgraph N and hyperparameter alpha. This should be declared in the paper and the scenario that the proposed method is not suitable can also be discussed.
>
> N denotes the (mini-)batch size, which is the higher the better and mostly decided by the GPU memory. Our results in Figure 2 and Table 3 indicate that our method is less sensitive to reduced batch size and consumes less memory compared to the recent baseline GraphCL. This allows our method to be applied to more scenarios. Regarding the hyper-parameter alpha, we empirically observed that the performance of our model is not sensitive to the value of alpha within a certain range of 1~10. Hence it does not limit the application scenario of LaGraph compared to other SSL methods. However, our method may have some limitations when learning from unattributed graphs. In this case, all information of the graph is contained in the topology of the graph, and our method may be less capable of fully adopting all the topological information in self-supervision. We have included detailed discussions regarding limitations in Appendix H.

---

> ### Author Response · Authors · 2021-11-23
> **Authors' follow-up on comments by Reviewer X3u8**
>
> Thank you again for your valuable comments! We believe we have addressed your concerns in our previous responses. We hope that you could consider updating your score if we do have addressed your concerns. Also, please let us know if there are any additional concerns or feedback. Thank you!

---

> > ### Comment · Reviewer_X3u8 · 2021-11-29
> > **Thanks for all the detailed responses from the authors.**
> >
> > I decide to keep my current score for the following two reasons: (1) the authors have addressed most of my previous concerns.
> > (2) After checking the comments from other reviewers, especially Reviewer t1HQ, I realize the deficiency of the novelty of this work in the field of Graph.
> >
> > Although I still think the idea of the latent graph is interesting, the connection of the proposed method with Graph learning needs to be further explored.

---

> > > ### Author Response · Authors · 2021-11-29
> > > **Re: Thanks for all the detailed responses from the authors.**
> > >
> > > Dear reviewer,
> > >
> > > Thank you for your further response. Regarding the novelty, we argue that LaGraph and existing methods are totally different methods and have substantial differences summarized below:
> > > - Theoretically, LaGraph and existing methods are **derived from different and independent theoretical grounding**. In particular, LaGraph objectives are derived as upper bounds of supervised objectives for latent graph prediction whereas BGRL is proposed as a variation of contrastive approaches, and, though sharing some similarity in formulations, consistency regularization is an independent technique proposed for GAN or self-training based on other different groundings.
> > > - For node-level tasks, BGRL computes its objective on all node embeddings, whereas the invariance term of LaGraph is **only computed on embeddings of masked nodes**. This is **due to the rigorous derivation of the upper bound**. Intuitively, the invariance term enforces the encode to capture less information from the node itself and more contextual information. If we also include unmasked nodes (as BGRL does), the invariance will lead to a contradicted effect, i.e., enforcing encoders to capture less information from contextual nodes, as it lets the representation remain consistent when its neighbor node (who is masked) is changed.
> > > - On implementation, LaGraph **does not require engineering tricks** such as stop gradient, EMA, and normalizations at final representations. The only requirements are the injective readout function for graph-level tasks and Lipschitz continuous neural networks to let the theorems hold.

---

### Official Review · Reviewer_t1HQ · 2021-11-06

**Correctness:** 3
**Technical Novelty And Significance:** 3
**Empirical Novelty And Significance:** 3
**Recommendation:** 5
**Confidence:** 3

**Main Review:**

Strengths:
* Simple architecture, with good results (Table 1)
* Theory is provided to justify their method and approach.
* The authors do a good job at explaining the conceptual differences between LaGraph and prior work.
* Comparisons with other SOTA methods on a number of graph-level and node-level benchmarks.

Weaknesses:
* It appears that the BGRL accuracies shown in Table 2 were reported in the original work and were not reproduced by the authors. The evaluation protocol used for LaGraph is different from the one used in BGRL: in BGRL, logistic regression is performed over the last layer of the encoder whereas in LaGraph (Appendix C), it is performed over the concatenation of the original features and the embeddings of the last layer. This is especially relevant for the transductive tasks, where the differences in accuracy are within 0.1% (which lies within the confidence interval). The authors should compare their methods using the same evaluation protocol.
* In table 2, LaGraph is only better than BGRL on one dataset (PPI). It would be interesting to see how this method provides additional benefits beyond performance since the methods give similar performance.
* The ablation studies are lacking as well. Given the similarity between LaGraph and BGRL, the ablations and other comparisons should include BGRL to show the performance gains.
* The assumption that each node embedding is sampled independently from the rest is not fully justified. The theory lacks any discussion of properties of the graph or its connectivity and thus is limited in its impact.
* While the authors describe the method as a new approach for latent graph prediction, the method is very similar to other predictive methods, especially BGRL and Graph BarlowTwins: the node representations of two views (view 1 = original graph, view 2 = graph with node dropout) are compared. The only difference is in the way the collapse is avoided is through the reconstruction term. Thus the novelty of the approach is limited in this respect.


**Summary Of The Paper:**

The paper proposes a new self-supervised learning framework for learning latent representations of graphs. Like recent SSL methods (BGRL,  Barlow twins), LaGraph does not rely on negative samples but only compares the embeddings of the graph to the embeddings of an augmented version of the graph. To avoid representational collapse, a reconstruction term (similar to denoising autoencoders) is used. They provide a theoretical analysis of the method which provides some connections to other approaches.

**Summary Of The Review:**

This paper introduces a new reconstruction-based loss for avoiding representation collapse and theory to justify their choice. In experiments on both graph- and node-level tasks, they show competitive performance with other SOTA methods. While the goal of providing some insights into SSL-based approaches is laudable, their theory is limited in that it doesn't provide any unique information that is graph specific and thus fails to provide insights into graph representation learning. When compared with similar SOTA methods, there is limited gain in performance; thus, it would be useful to evaluate other aspects of the algorithm and make stronger connections to the theory.

---

> ### Author Response · Authors · 2021-11-10
> **Response to comments by Reviewer t1HQ (con'd)**
>
> > The assumption that each node embedding is sampled independently from the rest is not fully justified. The theory lacks any discussion of properties of the graph or its connectivity and thus is limited in its impact.
>
> To be more clear, the assumption is that the observed node attributes are generated independently, **conditioned on its latent attribute**. It is absolutely possible that the attributes of the two observed nodes have dependencies. However, we assume that the dependent components are only contained in the latent attributes, without loss of generality (if the generation introduces dependencies, we can consider the dependent component together with the latent attributes as the new latent attributes). It is easier to understand the assumption in an image case. We consider a clean image as the latent data and observe its noisy version. We assume that the noise at each pixel is independent of each other (a common assumption in image denoising). If the noise is dependent on pixels, we call it a structural noise, such as a watermark on an image. In this case, we are still able to consider the watermark (structural noise) as a part of the image
>
> Regarding the limitations, as we focus on attributed graphs, our method does have some limitations with unattributed graphs in fully adopting topological information as the self-supervision. We have updated the draft to include detailed discussions on the limitations and potential solutions to tackle the limitations.
>
> > While the authors describe the method as a new approach for latent graph prediction, the method is very similar to other predictive methods, especially BGRL and Graph BarlowTwins: the node representations of two views (view 1 = original graph, view 2 = graph with node dropout) are compared. The only difference is in the way the collapse is avoided is through the reconstruction term. Thus the novelty of the approach is limited in this respect.
>
> > While the authors describe the method as a new approach for latent graph prediction, the method is very similar to other predictive methods, especially BGRL and Graph BarlowTwins: the node representations of two views (view 1 = original graph, view 2 = graph with node dropout) are compared. The only difference is in the way the collapse is avoided is through the reconstruction term. Thus the novelty of the approach is limited in this respect.
>
> In spite of the similarity in the objective of BGRL and the invariance term of the LaGraph objective, we argue that there have **key differences** between the invariance term in our objective and the BGRL/BT objectives. Regarding how the objective is computed, the reviewer may notice that **our invariance term is only computed for the masked nodes** in Theorem 1 and objective for node-level representations, in contrast to the BGRL/GT objective where all nodes are computed. It is worth noting that the objective is an upper bound only if the invariance is computed on the masked nodes, according to the derivation during Proof of Theorem 1. Intuitively, during the computation of a node representation, the invariance term enforces the encode to capture less information from the node itself and more contextual information. If we **also include unmasked nodes like BGRL, the invariance will lead to a contradicted effect**, i.e., enforcing encoders to capture less information from contextual nodes, as it lets the representation remain consistent when its neighbor node (who is masked) is changed. **Given the intuition and insights, how to adopt the invariance-related objectives (as well as consistency regularization) in SSL on graphs can be better guided in the future**.
>
> The similarity indicates a similar goal is achieved from different routes, i.e., empirical and theoretical, contrastive and predictive. To provide deeper insight into the similarities and differences, we include Section 3 and Appendix~D to analyze and discuss the connection between LaGraph and other methods. As mentioned in the above response, a benefit of theoretical grounding is the guidance in the design and implementation of the method and framework. To sum up, we believe that our approach is novel from the theoretical perspective and the insights provide enough contribution to the community studying SSL on graphs.

---

> ### Author Response · Authors · 2021-11-10
> **Response to comments by Reviewer t1HQ**
>
> We thank the reviewer for the valuable feedback and suggestions. Below are our item-wise responses to the comments.
>
> > It appears that the BGRL accuracies shown in Table 2 were reported in the original work and were not reproduced by the authors. The evaluation protocol used for LaGraph is different from the one used in BGRL: in BGRL, logistic regression is performed over the last layer of the encoder whereas in LaGraph (Appendix C), it is performed over the concatenation of the original features and the embeddings of the last layer. This is especially relevant for the transductive tasks, where the differences in accuracy are within 0.1% (which lies within the confidence interval). The authors should compare their methods using the same evaluation protocol.
>
> Self-supervised representation learning aims to obtain good representations from given unlabeled graph data. The representation can contain any information from the graph data as long as no knowledge about labels is used. For node-level tasks, we merge the output vector from the encoder and the original feature as the final learned representation. The intuition of this is based on the Bayesian rule where the learned encoder provides the prior knowledge [1] of data distribution whereas the given graph data serves as the observed samples. And the posteriori should be based on a combination of the priori (encoder output) and the observed data itself [2]. In addition, it is very common for SSL methods to combine information from multiple views of a graph as the final representation. For example, MVGRL merges the representations of the original graph and a diffused variation computed by two separate encoders as the final representation of the graph [3].
>
> From the experimental perspective, we admit that BGRL is a quite strong baseline method for transductive tasks. As the results are already on the same level as the performance of supervised training, it is very difficult to further obtain significant improvements. However, we argue that the non-significant performance boost on some transductive datasets does not degrade our main conclusion about the effectiveness of our method.
>
> We have updated our draft (in appendix E and H, due to limited space) to include the above discussion about the intuition of representation computing and the performance comparison. In addition, we are working on an ablation study that no concatenation is adopted for a deeper understanding of the proposed method. We will update the results once they are available.
>
> [1] Deep Image Prior. CVPR 2018.
>
> [2] High-Quality Self-Supervised Deep Image Denoising. NeurIPS 2019.
>
> [3] Contrastive Multi-View Representation Learning on Graphs. ICML 2020.
>
>
> > In table 2, LaGraph is only better than BGRL on one dataset (PPI). It would be interesting to see how this method provides additional benefits beyond performance since the methods give similar performance.
>
> As mentioned in the above response, the similar performance in the transductive tasks is due to the saturation in performance as BGRL has already achieved the same level of performance as supervised learning. However, when it comes to inductive tasks, where there is still a significant gap between the performance of BGRL and supervised learning, our method is able to bring significant improvements in performance.
>
> In addition to the performance, the proposed LaGraph has several advantages compared to BGRL. First, the proposed LaGraph is derived from a solid theoretical grounding that provides guidance to the designing and implementation of the method, such as an injective readout function. In contrast, BGRL requires several carefully designed implementation mechanisms, such as stop gradient, EMA, and batch normalization at the last layer, based on empirical results. Also, although not observed empirically, the potential collapsing issue is not theoretically guaranteed and is still possible to occur in some extreme cases. Second, the presence of theoretical grounding allows us to study the connections and differences between LaGraph and existing methods (**see more discussion in our response to the last comment**), which provide further insights into the study of self-supervised learning on graphs.
>
>
> > The ablation studies are lacking as well. Given the similarity between LaGraph and BGRL, the ablations and other comparisons should include BGRL to show the performance gains.
>
> The source code for BGRL was unavailable at the time of our submission and we hence did not include BGRL in the comparison. However, we are now working on getting the results and will update them here once we obtain them.

---

> > ### Comment · Reviewer_t1HQ · 2021-11-18
> > **Ablations of original features in evaluation**
> >
> > Thanks for your reply.
> >
> > To better understand how this difference in evaluation protocol impacts the final performance, it would be useful to see an ablation of LaGraph with only the representations learned by the model (without concatenated raw features). This would be a more fair comparison against the other models (GRACE, BGRL, MVGRL) that don't use this trick. To be very rigorous, you could also consider the performance of your other benchmark models when you append the same extra features as LaGraph.

---

> > > ### Author Response · Authors · 2021-11-20
> > > **Results for the ablation**
> > >
> > > Dear reviewer,
> > >
> > > Thank you for your further feedback! We have conducted the ablation for LaGraph on w/w.o. concatenation as suggested. Below are the results.
> > >
> > > |Dataset|Am. Comp.|Am. Pht.|Co. CS|Co. Phy|PPI|Flickr|Reddit|
> > > |---|---|---|---|---|---|---|---|
> > > |With concat|88.0±0.3|**93.5±0.4**|**93.3±0.2**|**95.8±0.1**|74.6±0.0|51.3±0.1|**95.2±0.0**|
> > > |W/o concat|**88.8±0.3**|92.7±0.4|92.6±0.2|95.3±0.1|**75.2±0.0**|**51.6±0.1**|94.8±0.0|
> > >
> > > When the concatenation is removed for node datasets, there is reduced performance for 4 of the 7 datasets but they are still on the same level with the performance of supervised learning. For the rest 3 datasets including the most challenging dataset PPI, there is a performance gain when removing the concatenation. Hence we believe that the conclusion still holds that, for node-level tasks, LaGraph provides additional performance gain on challenging datasets and is on par with supervised and BGRL performance on less challenging datasets.
> > >
> > > As the official implementation of BGRL is not publicly available yet, and we’re unable to reproduce the original results shown in the paper (even when using a public 3rd party implementation), a similar ablation for BGRL is currently unavailable. We will conduct the ablation in the future when the official code for BGRL is released.

---

> ### Author Response · Authors · 2021-11-23
> **Authors' follow-up on comments by Reviewer t1HQ**
>
> Thank you again for your valuable comments! We believe we have addressed your concerns in our previous responses. We hope that you could consider updating your score if we do have addressed your concerns. Also, please let us know if there are any additional concerns or feedback. Thank you!

---

### Official Review · Reviewer_Fi7g · 2021-11-07

**Correctness:** 3
**Technical Novelty And Significance:** 3
**Empirical Novelty And Significance:** 2
**Recommendation:** 6
**Confidence:** 3

**Main Review:**

+Strengths
  1. Good overall presentation of the technical content
  2. New contributions of tackling the challenges of conditional distribution $p(G|{G_I})$ instead of simply applying the noisy data reconstruction
  3. Sound theoretical analysis
  4. Comprehensive comparison with related work based on various techniques, e.g., denoising autoencoders, information bottleneck principle, and local-global mutual information maximization for contrastive learning

+Weaknesses
  1. Some problem definitions, preliminaries, and details regarding the proposed method are not very clear (see Detailed Comments 1)
  2. Some details regarding the experiments are not given (see Detailed Comments 2)
  3. No discussion about the inherent limitations of the proposed method and possible solutions (see Detailed Comments 3)

+Details Comments
1. Some problem definitions and technical details regarding the proposed method are unclear and need to be further clarified.

  For the problem definitions in Section 2.1, the authors claim that they 'consider an undirected graph'. It seems the proposed method can only derive the representations of one single graph. However, in Section 2.4, the authors claim that they consider a mini-bath of $N$ graphs, which is inconsistent with that in Section 2.1. In the experiments, the author applied the proposed method to multiple graphs for graph classification, which is also inconsistent with that in Section 2.1 (i.e., learning representation of one single graph). It is better to formulate the graph representation learning based on a set of graphs (but not a single graph) in Section 2.1.

  It is also recommended to give formal definitions regarding transductive and inductive graph representation learning. Especially for the inductive graph embedding, it is unclear that the authors consider inductive node-level task for new unseen nodes of a graph or across multiple graphs (w.r.t. what they conduced in the experiments). The authors should also  verify some constraints of inductive graph embedding, e.g., different graphs can have different node sets, all the graphs must have the same feature dimensionality, etc.

  In the first paragraph of Section 2.2, the authors claim that the latent data ${\bf{x}}_{I}$ determines the semantic of the observed data ${\bf{x}}$. What does the 'semantic' of observed data (e.g., in terms of graphs) mean? It is better to give some simple examples to explain the 'semantic' of an observed graph.

  In the 2nd paragraph in Section 2.2, the authors claim that 'the pair of graphs have matched structure and feature dimensions'. What does 'matched structure' mean? From my perspective, one may have ambiguous understandings. It may indicate that the two graphs (i.e., an observed graph and its latent graph) have the same node set (but with different topology described by different edge sets). It can also indicate that the two graphs have the same node set and edge set. Until I read Theorem 1 with the notations $G=({\bf{A}}, {\bf{X}})$ and $G_I=({\bf{A}}, {\bf{F}})$, I know that it indicate the latter case. It is better to give the formal definitions of the observed graph and its latent graph at the beginning of Section 2.2.

  In Corollary 1 and Corollary 2, ${\bf{H}}'$ denotes the embedding of 'masked graph', but what is a 'masked graph'? Is it with the same definition as 'latent graph'? It seems there is no definition of 'masked graph' before Corollary 1 and Corollary 2.

  In Section 2.4, what is the definition of ${\bf{X}}_{(i,J_i^c)}$? Is it a single masked variant of an observed graph? If so, the proposed method only generated one masked variant for each graph. It is not clear how many mask variants are generated for each single observed graph $G_i$. In fact, some details of the proposed method can be clearly presented using the pseudo-code (even in the appendix) but there is no pseudo-code to describe the overall training and inference procedures of the proposed method.

  From my perspective, Fig. 1 does not accurately present the basic ideas of the proposed method. In this paper, the authors consider the representation learning of attributed graph, where each node is associated with a feature vector. However, node feature vectors are not presented in Fig. 1. For the masked graph in the 'Input graphs' subfigure, some of the nodes are crossed. What are the crossed nodes? Are they the nodes selected to be masked? The masking operation is conducted on node features but not both on topology and attributes. The current presentation of the masked graph implies that one needs to first remove the crossed nodes from the original graph (i.e., the masked graph is then with the node set $J^c$), which is inconsistent with the proposed method. Moreover, in the 'Representation' subfigures of Fig. 1, there is the topology of two graphs, which are not the derived representations. In fact, the representations are low-dimensional vectors but not graph topology as presented in the 'Representation' subfigures. Why the graph-level representations of the observed graph and latent graph are denoted as ${\bf{Z}}$ and ${\bf{Z}}'$ in bold capital letters in Fig. 1? Usually, bold capital letters are used to describe matrices. From my perspective, they should be presented in bold lowercase letters, i.e., ${\bf{z}}$ and ${\bf{z}}'$ as in Eq. (6), which indicate that they are vectors (but not matrices).

  In Section 3.1, the author claim that the proposed upper bounds 'allow an encoder to access a certain level of information of the masked nodes, whose representations can be as good as ones from supervised learning'. I am curious about how the information of the masked nodes is captured by the proposed method and why the derived representations can be as good as ones from supervised learning? It is better to give some simple examples to help the readers better understand this point.

2. Some details regarding the experiments are missing. Some experiment settings also need further clarification.

  There are no statistic details regarding the datasets used in the experiments, e.g., number of graphs, number of nodes, number of edges, dimensionality of node attributes, etc. Details regarding the evaluation protocols for different tasks are also not given. Although the authors claim that they follow the experiment settings of prior work, some important content (e.g., experiment settings) of a paper should be self-contained and do not rely on other papers. Hence, it is recommended to briefly introduce the statistic details and evaluation protocols even in the appendix.

  As I mentioned in the Detailed Comments 1, what does the inductive node-level classification indicate? Does it refer to the (i) classification for new unseen nodes in a graph, (ii) classification across multiple graphs, or (iii) both of them? It is better to give a clear definition of this task.

  In the graph classification task, how node2vec and sub2vec are used to derive the graph embedding? As I know node2vec and sub2vec can only derive the vector representations for each node and subgraph, respectively. How did the authors convert the node and subgraph embeddings to the graph embedding? How did the subgraphs sampled for sub2evc? The settings of other baselines should also be briefly introduced.

  Why did the authors use node2vec in graph classification but use DeepWalk in transductive node classification? As I know, node2vec and DeepWalk are two node embedding baselines with similar motivations. Why not use both of them in graph classification and transductive node classification?

  In Table 2, why did the authors use different evaluation metrics for transductive and inductive node classification (i.e., accuracy and Micro-average F1 score)?

  In Table 1 and Table 2, why some of the results of GL (e.g., on NCI1, PROTEINS, and DD), WL (e.g., on DD and COLLAB), DGK, Node2Vec, Sub2Vec, Graph2Vec, BGRL-GCN, and BGRL-GAT are not given?

  For the empirical analysis regarding the batch size and number of sampled nodes in Fig. 2 and Table 3, besides the memory usage, what is the corresponding training time w.r.t. each setting?

3. The limitations of the proposed method and possible solutions are not fully discussed in Section 5. A good paper should also comprehensively discuss its limitations in addition to its contributions.

  From my perspective, the proposed method is designed for attributed graphs and relies heavily on the reconstruction of node features. How can the proposed method deal with the representation learning on graphs without any available node attributes? By reading the appendix, I know that the authors tried to apply the one-hot vector of degree for each node as the node attributes. It is not an elegant way for the inductive representation learning across multiple graphs, where the model can only be trained on some of the graphs (i.e., the training set) and be generalized to the other unseen graphs (i.e., the test set). During the training, only the information of training set is given, which indicates that the node degree of the test set is unknown. When the node degrees in the test set have a wide value range but those in the training set only have a small value range, it is possible that most of the node attributes in a test graph are zero vectors. Moreover, what about the graph embedding on a set of regularized graphs without node attributes, where all the nodes have the same degree value? In this case, all the node feature vectors are the same.

  In addition to reconstructing the node attributes, why not consider another direction of reconstructing the topology? Namely, we can derive an observed graph and its latent graph with different set of edges (but with the same node attributes). The authors can give some discussions regarding this direction.

  To scale a GNN to large graphs (in terms of the number of nodes) is a hot topic in recent research. The authors also mention that the proposed method can be applied to large graphs. However, as I can check from the descriptions of experiments, the largest graph (i.e., Reddit) used in the experiments only has 232,965 nodes, which cannot be considered as a very large graph. For the reported evaluation results in Table 2, it seems that the authors did not apply any mini-batch subgraph sampling strategies. For large graphs (e.g., with more than one million nodes), most of the GNN structures need certain mini-batch subgraph sampling strategies. How the sampling strategies can be applied to the proposed method considering both the effects of topology and attributes? The authors can give some further discussions regarding this point.

  Most of the SSL-based graph embedding methods are based on unsupervised training loss and evaluated via supervised downstream tasks (e.g., node and graph classification). I am curious about their ability to deal with some unsupervised downstream tasks (e.g., disjoint and overlapping community detection).

**Summary Of The Paper:**

  In this paper, the authors propose a novel SSL method (i.e., LaGraph) for graph representation learning based on latent graph prediction. The overall presentation is good with sound theoretical analysis. There is also a theoretical comparison between LaGraph and various related work with different categories. Extensive experiments were conducted on various datasets for both node-level and graphs-level tasks, where the proposed method has competitive performance to a set of baselines. However, there are some details regarding the problem definitions, technical content, and experiment settings that need further clarification.

**Summary Of The Review:**

This paper has good overall presentation of the technical content and sound theoretical analysis regarding the proposed method as well as related work. From my perspective, the proposed method is novel. However, there are some details regarding the problem definitions and experiment settings that need to be further verified.

---

> ### Author Response · Authors · 2021-11-10
> **Response to detailed comments 3 (con'd)**
>
> > To scale a GNN to large graphs (in terms of the number of nodes) is a hot topic in recent research. The authors also mention that the proposed method can be applied to large graphs. However, as I can check from the descriptions of experiments, the largest graph (i.e., Reddit) used in the experiments only has 232,965 nodes, which cannot be considered as a very large graph. For the reported evaluation results in Table 2, it seems that the authors did not apply any mini-batch subgraph sampling strategies. For large graphs (e.g., with more than one million nodes), most of the GNN structures need certain mini-batch subgraph sampling strategies. How the sampling strategies can be applied to the proposed method considering both the effects of topology and attributes? The authors can give some further discussions regarding this point.
>
> Results in Table 2 do not apply mini-batched training to perform a fair comparison with the baseline methods, who perform training on the entire graph too. However, we conduct additional studies in Sec 4.2 (Training on Sub-Graphs for Large-Scale Datasets) to show the robustness of our method when a subgraph (mini-batch of node) is used for each iteration of training. We believe it can indicate the performance of our method with subgraph (mini-batch) sampling on even larger datasets.
>
> > Most of the SSL-based graph embedding methods are based on unsupervised training loss and evaluated via supervised downstream tasks (e.g., node and graph classification). I am curious about their ability to deal with some unsupervised downstream tasks (e.g., disjoint and overlapping community detection).
>
> Our experiments follow the baseline methods to perform evaluations on commonly used tasks in the self-supervised learning studies for graphs. The study of self-supervised representation learning on unsupervised downstream tasks is indeed an interesting topic but can fall into a separate work. We thank the reviewer for suggesting the interesting direction and will definitely conduct further studies in it. However, due to the time constraints, we are unable to obtain the results during the discussion session.

---

> ### Author Response · Authors · 2021-11-10
> **Response to detailed comments 3**
>
> >3. The limitations of the proposed method and possible solutions are not fully discussed in Section 5. A good paper should also comprehensively discuss its limitations in addition to its contributions.
>
> There are indeed several limitations as the reviewer mentioned. We have included an additional section in Appendix H to discuss the limitations and their potential solutions. Below are our item-wise responses.
>
> > From my perspective, the proposed method is designed for attributed graphs and relies heavily on the reconstruction of node features. How can the proposed method deal with the representation learning on graphs without any available node attributes? By reading the appendix, I know that the authors tried to apply the one-hot vector of degree for each node as the node attributes. It is not an elegant way for the inductive representation learning across multiple graphs, where the model can only be trained on some of the graphs (i.e., the training set) and be generalized to the other unseen graphs (i.e., the test set). During the training, only the information of the training set is given, which indicates that the node degree of the test set is unknown. When the node degrees in the test set have a wide value range but those in the training set only have a small value range, it is possible that most of the node attributes in a test graph are zero vectors. Moreover, what about the graph embedding on a set of regularized graphs without node attributes, where all the nodes have the same degree value? In this case, all the node feature vectors are the same.
>
> Although we mainly focus on graphs with attributed nodes (as stated in Section 2.1), there do exist cases when node attributes are unavailable. It is a common approach, especially for graph-level tasks, to adopt node degree as its attributes [4]. We are aware of the problem that “node degrees in the test set have a wide value range but those in the training set only have a small value range”. That’s why we include the degree threshold, i.e., we consider the degree as k when it is greater than k, on some datasets to tackle the potential inconsistency issue in node degree distributions between training and testing graphs.
>
> The regular graphs (whose node degrees are the same) are extreme cases that our current solution fails. However, for these cases, the nodes (or graphs) are indistinguishable by even GCN and GIN trained under supervised learning,  and more advanced GNNs are required [5]. A feasible alternative for the case (if the encoder is capable) is to replace the reconstruction of node attributes with the reconstruction of the connectivity of a node i.e., to reconstruct the adjacency matrix with a prediction head consisting of MLP and dot production between node embeddings.
>
> [4] ​​How Powerful Are Graph Neural Networks. ICLR 2019.
>
> [5] Distance Encoding: Design Provably More Powerful Neural Networks for Graph Representation Learning. NeurIPS 2020.
>
> > In addition to reconstructing the node attributes, why not consider another direction of reconstructing the topology? Namely, we can derive an observed graph and its latent graph with a different set of edges (but with the same node attributes). The authors can give some discussions regarding this direction.
>
> The reconstruction of topology described by the reviewer is indeed a promising direction. Although it does not fit in our current theoretical framework, it is possible to derive a similar framework for topology reconstruction (e.g., based on an upper bound of link prediction objective) following the similar idea to further boost the model performance by capturing the topological information.

---

> ### Author Response · Authors · 2021-11-10
> **Response to detailed comments 2 (con'd)**
>
> > Why did the authors use node2vec in graph classification but use DeepWalk in transductive node classification? As I know, node2vec and DeepWalk are two node embedding baselines with similar motivations. Why not use both of them in graph classification and transductive node classification?
>
> We use the results of Node2Vec from [1] on graph classification tasks. However, there is no implementation of DeepWalk on graph classification tasks. On the other hand, the DeepWalk is the most commonly used traditional baseline of SSL methods for node-level tasks, whereas node2vec is seldomly used. We follow previous methods to show only results of DeepWalk for transductive node-level tasks. For inductive tasks, as there are more important SSL baseline results, we hence omit the result of DeepWalk due to page limit. However, the results are available below.
>
> |PPI | Flickr | Reddit|
> |---|---|---
> |52.9 | 27.9 | 32.4|
>
> > In Table 2, why did the authors use different evaluation metrics for transductive and inductive node classification (i.e., accuracy and Micro-average F1 score)?
>
> The two evaluation metrics are used by previous baseline methods. We follow previous methods for a consistent comparison.
>
> > In Table 1 and Table 2, why some of the results of GL (e.g., on NCI1, PROTEINS, and DD), WL (e.g., on DD and COLLAB), DGK, Node2Vec, Sub2Vec, Graph2Vec, BGRL-GCN, and BGRL-GAT are not given?
>
> Results of kernel-based methods (GL, WL) and traditional methods (x2vec) are from previous works [2,3]. All missing results of kernel-based methods and missing results of traditional methods on DD and COLLAB are due to out-of-memory issues. Results of node2vec on the remaining 3 datasets are unavailable due to unclear implementation [1]. Missing results of BGRL are because the official code of BGRL is not publicly available yet and currently the 3rd party implementation does not reproduce original results on existing dataset. We will complete the table in the future when the official code is released.
>
> [2] InfoGraph: Unsupervised and Semi-supervised Graph-Level Representation Learning via Mutual Information Maximization. ICLR 2020.
>
> [3] Graph Contrastive Learning with Augmentations. NeurIPS 2020.
>
> > For the empirical analysis regarding the batch size and number of sampled nodes in Fig. 2 and Table 3, besides the memory usage, what is the corresponding training time w.r.t. each setting?
>
> We run LaGraph and GraphCL with the same batch size of 128 for 20 epochs. Below is the average time cost in seconds per epoch. The exact total numbers of epochs are based on grid-search for both methods but are at the same level.
>
> |Method|NCI1|PROTEINS|DD|MUTAG|COLLAB|RDT-B|RDT-M5K|IMDB-B|
> |---|---|---|---|---|---|---|---|---|
> |LaGraph|20.937|0.946|98.276|0.173|6.054|3.371|16.702|0.676|
> |GraphCL|25.485|1.962|163.764|0.256|137.815|83.769|264.231|1.355|
>
> We observe a significantly longer training time for GraphCL on COLLAB, RDT-B, and RDT-M5K, which may be due to more time spent on performing augmentations on graphs with more edges.

---

> > ### Comment · Reviewer_Fi7g · 2021-11-21
> > **Regarding the presentation of experiment results**
> >
> > Thanks for the response to my comments. According to my understanding, the authors directly show some of the experiment results reported in other related papers for different tasks. Although it is a widely-used strategy, in which the authors do not need to run other baselines by themselves, it is not a recommended setting from my perspective, especially when the authors need to consider multiple tasks and some of the previous papers do not provide experiment results for some of the tasks, datasets, or baselines. I think this is why some of the results (e.g., Nodev2d on DD) are not given and the evaluation metrics for transductive and inductive node classification are different.
> >
> > For some blank results (e.g., Nodev2d on DD, Micro-average F1 score for transductive node classification, etc.), the authors can still run the experiments by themselves and clearly denote the resources of different experiment results (e.g., from a paper or conducted by themselves). Some papers also adopt such a strategy. I understand that it would be very time-consuming for the authors to fill all the blank results and cannot finish before the revision period. I will consider such a fact when making the final decision.

---

> > > ### Author Response · Authors · 2021-11-21
> > > **Re: Regarding the presentation of experiment results**
> > >
> > > Thank you for your further comments. We agree that completed results of earlier methods with detailed descriptions are important and would help future studies to perform solid evaluations. We will work on it to obtain as many results as possible and update them in the final version.

---

> > > ### Author Response · Authors · 2021-11-23
> > > **Authors' follow-up and more results available**
> > >
> > > Dear Reviewer Fi7g,
> > >
> > > Below are the previously missing results we currently obtain for the baseline methods x2vec. Results for kernel methods will be updated in the final version.
> > >
> > > |Methods|DD|COLLAB|
> > > |---|---|---|
> > > |Sub2Vec|73.6±1.5|62.1±1.4|
> > > |Graph2Vec|76.2±0.1|59.9±0.0|
> > >
> > > |Methods|DD|COLLAB|RDT-B|RDT-M5K|IMDB-B|
> > > |---|---|---|---|---|---|
> > > |Node2Vec|75.1±0.5|55.7±0.2|73.8±0.5|34.07±0.4|50.0±0.8|
> > >
> > > For all tables, our primary goal is to compare LaGraph with self-supervised GNN baselines, and results for kernel and walk-based methods are shown as references. Hence we respectfully point out that the currently included methods are sufficient to support our conclusions. While we would like to make the tables as complete as possible, we believe that it is NOT a key factor for the evaluation of our work.
> > >
> > > Besides this, we believe we have addressed your concerns in our previous responses. We hope that you could consider updating your score if we do have addressed your concerns. Also, please let us know if there are any additional concerns or feedback. Thank you again for your valuable comments!

---

> ### Author Response · Authors · 2021-11-10
> **Response to detailed comments 2**
>
> > 2. Some details regarding the experiments are missing. Some experiment settings also need further clarification.
>
> > There are no statistic details regarding the datasets used in the experiments, e.g., number of graphs, number of nodes, number of edges, dimensionality of node attributes, etc. Details regarding the evaluation protocols for different tasks are also not given. Although the authors claim that they follow the experiment settings of prior work, some important content (e.g., experiment settings) of a paper should be self-contained and do not rely on other papers. Hence, it is recommended to briefly introduce the statistic details and evaluation protocols even in the appendix.
>
> We include the dataset statistics in Table 4 in Appendix E as suggested. Regarding the evaluation protocols, we describe how they are performed in the second paragraph of Section 4, quoted “In particular, for both levels, we first train the graph encoder on unlabeled graph datasets with the corresponding self-supervised objective. We then compute and freeze the corresponding representations and train a linear classification model on top of the fixed representations with their corresponding labels. Linear SVM and the regularized logistic regression are employed as linear classifiers for graph-level datasets and node-level datasets. For inductive node-level datasets, the self-supervised training is only performed on graphs in the training datasets whereas the test graphs are unavailable during the self-supervised training.”
>
> > As I mentioned in the Detailed Comments 1, what does the inductive node-level classification indicate? Does it refer to the (i) classification for new unseen nodes in a graph, (ii) classification across multiple graphs, or (iii) both of them? It is better to give a clear definition of this task.
>
> For the PPI dataset consisting of 24 graphs, the training and testing nodes are split by graphs. In other words, node representations are learned from training graphs, and the encoder is evaluated on testing graphs i.e., (ii). For Flickr and Reddit each consisting of only one graph, the training and testing nodes are from the same graph. During self-supervised training all test nodes are masked-out. During evaluation, all training nodes are masked-out. So it is (i). For both cases, no testing node is used during the self-supervised training stage and no training node is used during evaluation.
>
> > In the graph classification task, how node2vec and sub2vec are used to derive the graph embedding? As I know node2vec and sub2vec can only derive the vector representations for each node and subgraph, respectively. How did the authors convert the node and subgraph embeddings to the graph embedding? How did the subgraphs sampled for sub2evc? The settings of other baselines should also be briefly introduced.
>
> According to the original paper, Sub2Vec learns “the embedding of each graph by treating them as a subgraph of a union of all the graphs”. As for Node2Vec, the results are from [1], whose implementation details are not clear. However, it is common to compute a graph representation from node embeddings by performing a readout such as global sum or average pooling.
>
> [1] graph2vec: Learning Distributed Representations of Graphs

---

> ### Author Response · Authors · 2021-11-10
> **Response to detailed comments 1 (con'd 2)**
>
> > In Section 3.1, the authors claim that the proposed upper bounds 'allow an encoder to access a certain level of information of the masked nodes, whose representations can be as good as ones from supervised learning'. I am curious about how the information of the masked nodes is captured by the proposed method and why the derived representations can be as good as ones from supervised learning? It is better to give some simple examples to help the readers better understand this point.
>
> The upper bounds consist of two components. Considering the representation of a node, the reconstruction term enforces information of the node itself to be preserved in the representation (otherwise the node attributes cannot be reconstructed). In addition, the invariance term ensures that the node representation does not fully depend on the node itself, but also depends on the contextual nodes. Only including one of them will lead to that the representation captures too much or not enough information from the original node features.
>
> Taking the denoising autoencoder as an example, it masks some input nodes and reconstructs the node features from its neighbors in the masked graph. As the node itself is noisy (masked), the encoder is only trained to capture contextual information, excluding the center node itself when computing its representation. This leads to a discrepancy to how an encoder performs during a supervised learning, where both contextual information and center node information are captured.
>
> Due to the space limit, we will reorganize the space of the main text and include more explanations regarding this statement in the final version.

---

> ### Author Response · Authors · 2021-11-10
> **Response to detailed comments 1 (con'd)**
>
> > In the first paragraph of Section 2.2, the authors claim that the latent data determines the semantic of the observed data . What does the 'semantic' of observed data (e.g., in terms of graphs) mean? It is better to give some simple examples to explain the 'semantic' of an observed graph.
>
> The “semantic” of observed data refers to any meaningful information in the data related to its identity and properties. Similarly to images, the observed graph data may include “noise” during the measurement and collection of data, which noise is assumed irrelevant to its semantic. For example, in commodity-buyer graphs used in recommendation systems, a commodity may be mis-categorized by the merchant (“noise” in node attributes) but the mis-categorization should not change the identity of the commodity. In molecular graphs, changing the attributes of certain atoms or even replacing atoms does not affect its biomedical or chemical properties (semantic) of the molecule.
>
> > In the 2nd paragraph in Section 2.2, the authors claim that 'the pair of graphs have matched structure and feature dimensions'. What does 'matched structure' mean? From my perspective, one may have ambiguous understandings. It may indicate that the two graphs (i.e., an observed graph and its latent graph) have the same node set (but with different topology described by different edge sets). It can also indicate that the two graphs have the same node set and edge set. Until I read Theorem 1 with the notations G=(A,X) and G_I=(A,F), I know that it indicates the latter case. It is better to give the formal definitions of the observed graph and its latent graph at the beginning of Section 2.2.
>
> The reviewer’s understanding is correct. The two graphs have the same node set and edge set, but differ in the node attributes. We have updated Section 2.2 to make it more clear.
>
> > In Corollary 1 and Corollary 2, H’ denotes the embedding of 'masked graph', but what is a 'masked graph'? Is it with the same definition as 'latent graph'? It seems there is no definition of 'masked graph' before Corollary 1 and Corollary 2.
>
> The masked graph is denoted by (A, X_{J^c}). All attributes of a certain portion of nodes are masked by random noise, which is described in the second paragraph, i.e.,  X_{J^c}:=\mathds{1}_{J^c}\odot X+\mathds{1}_{J}\odot M.
>
> > In Section 2.4, what is the definition of X_{(i, J_{ic})}? Is it a single masked variant of an observed graph? If so, the proposed method only generated one masked variant for each graph. It is not clear how many mask variants are generated for each single observed graph G_i. In fact, some details of the proposed method can be clearly presented using the pseudo-code (even in the appendix) but there is no pseudocode to describe the overall training and inference procedures of the proposed method.
>
> X_{(i, J_{ic})} denotes the masked variant of the i-th (observed) graph. The mask is randomly generated for each graph at each iteration during training. Whether a node is masked is generated from a Bernoulli distribution and the masking values are generated from a Gaussian distribution. As suggested, we have provided the Pseudo-code for computing LaGraph objectives in Appendix C.
>
>
> > From my perspective, Fig. 1 does not accurately present the basic ideas of the proposed method. In this paper, the authors consider the representation learning of attributed graphs, where each node is associated with a feature vector. However, node feature vectors are not presented in Fig. 1. For the masked graph in the 'Input graphs' subfigure, some of the nodes are crossed. What are the crossed nodes? Are they the nodes selected to be masked? The masking operation is conducted on node features but not both on topology and attributes. The current presentation of the masked graph implies that one needs to first remove the crossed nodes from the original graph (i.e., the masked graph is then with the node set J_c), which is inconsistent with the proposed method. Moreover, in the 'Representation' subfigures of Fig. 1, there is the topology of two graphs, which are not the derived representations. In fact, the representations are low-dimensional vectors but not graph topology as presented in the 'Representation' subfigures. Why the graph-level representations of the observed graph and latent graph are denoted as Z and Z’ in bold capital letters in Fig. 1? Usually, bold capital letters are used to describe matrices. From my perspective, they should be presented in bold lowercase letters, i.e., z and z’ as in Eq. (6), which indicate that they are vectors (but not matrices).
>
> We thank the reviewer for pointing out the confusing parts. A node is crossed iff all attributes of the node are masked by random values. The topology of the graph is still reserved. We have updated the caption of Fig. 1 to make it more clear. Besides, the notations of graph representations have been fixed in the figure.

---

> ### Author Response · Authors · 2021-11-10
> **Response to detailed comments 1**
>
> We thank the reviewer for the thorough feedbacks, valuable suggestions and insights. Below are our item-wise responses to the comments. Certain parts of the draft are also updated as suggested.
>
> > For the problem definitions in Section 2.1, the authors claim that they 'consider an undirected graph'. It seems the proposed method can only derive the representations of one single graph. However, in Section 2.4, the authors claim that they consider a mini-bath of N graphs, which is inconsistent with that in Section 2.1. In the experiments, the author applied the proposed method to multiple graphs for graph classification, which is also inconsistent with that in Section 2.1 (i.e., learning representation of one single graph). It is better to formulate the graph representation learning based on a set of graphs (but not a single graph) in Section 2.1.
>
> The first paragraph of Section 2.1 intends to introduce notations used in the paper. However, we do realize that it may cause some confusion. We have updated the second paragraph (description of self-supervised learning) to make it clear that the self-supervised learning is performed on a set of graphs.
>
> > It is also recommended to give formal definitions regarding transductive and inductive graph representation learning. Especially for the inductive graph embedding, it is unclear that the authors consider inductive node-level task for new unseen nodes of a graph or across multiple graphs (w.r.t. what they conduced in the experiments). The authors should also verify some constraints of inductive graph embedding, e.g., different graphs can have different node sets, all the graphs must have the same feature dimensionality, etc.
>
> For node-level tasks, data typically consists of one or more large graphs with nodes and edges. The goal of self-supervised learning on such node-level datasets is to learn high-quality node representation that facilitates downstream node classification tasks. Our node-level experiments are divided into two categories, namely, transductive and inductive tasks.
>
> Transductive self-supervised learning of node representation allows utilization of all data at hand to pre-train GNNs for downstream tasks. Although labels of nodes are not visible during pre-training, patterns and information present in all nodes are observed. In contrast to transductive learning, inductive self-supervised learning only allows using a portion of data to pre-train GNNs, while holding out a certain amount of data for downstream tasks.
>
> ​​Our inductive tasks include two cases. (1), the PPI dataset consists of 24 graphs, and the training and testing nodes are split by graphs. In this case, the inductive task is considered across multiple graphs. In other words, node representations are learned from training graphs, and the encoder is evaluated on testing graphs.  (2), Flickr and Reddit each consist of only one graph, the training and testing nodes are from the same graph. During self-supervised training, all test nodes are masked-out. During evaluation, all training nodes are masked-out, i.e.,test nodes are unseen nodes of the graph. For both cases of inductive learning, data used during the self-supervised training stage and data used during evaluation stage are distinct, but the feature dimensionality should be the same for data used in both stages.
>
> We have included descriptions as well as the constraints of transductive and inductive learning in Appendix as suggested by the reviewer.

---

> ### Comment · Reviewer_Fi7g · 2021-11-29
> **Decision after rebuttal and revision**
>
> By Comprehensively considering the authors' response and revision, I decide to change my score from 5 to 6.
>
> From my perspective, it is not recommended using two different evaluation metrics for transductive and inductive node classification. The revision version still used different metrics for these two tasks, which relies heavily on the experiment results reported by prior papers. The authors cannot give some details of these baselines due to the unclear descriptions of these prior papers.
>
> There are still some unclear presentation in the revised paper. In Fig. 1, the node attributes (e.g., the original node attributes and masked attributes) are not accurately presented in the revised version. In the input graph, each node is associated with an attribute vector but there are no attribute vectors in Fig. 1. Moreover, how to produce the masked node attributes is not illustrated in Fig. 1. In Table 4, the feature dimensionality of RDT-B, RDT-M5K, and IMDB-B are 1. As I known these 3 datasets are sets of graphs without attributes. According to the descriptions of the authors, they use one-hot representation of node degree as the node features. From my perspective, the corresponding feature dimensionality should not be 1.
>
> Other reviewers also have some concerns regarding the novelty of the proposed method.

---

> > ### Author Response · Authors · 2021-11-29
> > **Re: Decision after rebuttal and revision**
> >
> > Dear reviewer,
> >
> > Thank you for the further response and update. Regarding the metrics for node-level tasks, we will include the F1 scores of our methods (and as many baselines as possible) for the transductive datasets in the final version so that future works can use our results under a consistent metric. We will also further update Figure 1 and dimensions in Table 4 to make them more clear in the final version.

---

### Author Response · Authors · 2021-11-20
**Summary of authors responses. Please let us know if any questions or additional comments**

Dear reviewers and area chair,

We thank all the reviewers for taking time to provide insightful comments and constructive suggestions. They are very helpful for us to strengthen the submission.

The reviewers agree that our method is impressive, theoretically sound, and provides insightful analyses, but concerns are raised by the reviewers. Regarding the concerns and questions, we have provided detailed responses and updated our draft accordingly to refine our presentation and address the concerns. Given your overall positive comments, we hope that you could update your evaluation if we have addressed your concerns.

To summarize, besides answering questions and refining our representation, we have 1) included more detailed discussions to highlight **both theoretical and intuitive difference** between LaGraph and BGRL (and consistency regularization) and discuss how the derivation of our objectives can provide guidance on how to better adopt the invariance-like objectives, 2) for the node-level experimental results, we highlight that LaGraph provides significant performance gain on the most challenging dataset and is on par with the supervised performance on datasets that are less challenging, 3) conducted additional ablation studies for a better understanding of our methods, 4) included an additional section in appendix to discuss the potential limitations of our methods, why they are not critical limitations, and solutions to address the limitations.

We would like to know if our response has properly addressed your concerns. Please let us know if you have any questions or additional comments.

---

### Decision · Program_Chairs · 2022-01-20

**Decision:**

Reject

**Comment:**

This paper studies self-supervised learning for graph neural networks by proposing a framework called LaGraph. Both theoretical analysis and experimental evaluation are provided in the paper.

We acknowledge the merits of this paper, which include studying a relatively less explored topic, providing theoretical analysis and comparison with other methods, and requiring less memory than a strong baseline.

On the other hand, there are also outstanding concerns (even after the discussions) regarding the novelty and significance of the proposed method (despite the claims of the authors during the discussions), whether the performance improvement over strong baselines is significant across different datasets, and missing a more comprehensive ablation study (beyond the preliminary results provided during the discussion period), among others.

In its current form, this is certainly a borderline paper for a top conference such as ICLR. It would be a better paper if the outstanding concerns could also be addressed before publication.